# Adapting from Low to High: An Update to CO_2_-Concentrating Mechanisms of Cyanobacteria and Microalgae

**DOI:** 10.3390/plants12071569

**Published:** 2023-04-06

**Authors:** Elena V. Kupriyanova, Natalia A. Pronina, Dmitry A. Los

**Affiliations:** K.A. Timiryazev Institute of Plant Physiology, Russian Academy of Sciences, 127276 Moscow, Russia

**Keywords:** *Chlamydomonas reinhardtii*, CO_2_-concentrating mechanism, C_3_ photosynthesis, cyanobacteria, microalgae, inorganic carbon transport, carbonic anhydrase, carboxysome, pyrenoid, Rubisco

## Abstract

The intracellular accumulation of inorganic carbon (C_i_) by microalgae and cyanobacteria under ambient atmospheric CO_2_ levels was first documented in the 80s of the 20th Century. Hence, a third variety of the CO_2_-concentrating mechanism (CCM), acting in aquatic photoautotrophs with the C_3_ photosynthetic pathway, was revealed in addition to the then-known schemes of CCM, functioning in CAM and C_4_ higher plants. Despite the low affinity of ribulose-1,5-bisphosphate carboxylase/oxygenase (Rubisco) of microalgae and cyanobacteria for the CO_2_ substrate and low CO_2_/O_2_ specificity, CCM allows them to perform efficient CO_2_ fixation in the reductive pentose phosphate (RPP) cycle. CCM is based on the coordinated operation of strategically located carbonic anhydrases and CO_2_/HCO_3_^−^ uptake systems. This cooperation enables the intracellular accumulation of HCO_3_^−^, which is then employed to generate a high concentration of CO_2_ molecules in the vicinity of Rubisco’s active centers compensating up for the shortcomings of enzyme features. CCM functions as an add-on to the RPP cycle while also acting as an important regulatory link in the interaction of dark and light reactions of photosynthesis. This review summarizes recent advances in the study of CCM molecular and cellular organization in microalgae and cyanobacteria, as well as the fundamental principles of its functioning and regulation.

## 1. Introduction

Accumulation of intracellular inorganic carbon (C_i_) in microalgae and cyanobacteria is one of the adaptive mechanisms available in photoautotrophs that allow them to perform efficient photosynthesis.

The central element of all types of photosynthetic carbon metabolism is the reductive pentose phosphate (RPP) cycle (Calvin-Benson-Bassham cycle, C_3_ cycle). Ribulose-1,5-bisphosphate carboxylase/oxygenase (Rubisco), the key enzyme of the RPP cycle, combines CO_2_ with ribulose-1,5-bisphosphate (RuBP). In many photoautotrophs, Rubisco is characterized by a rather low affinity for CO_2_ (*K*_M_ (CO_2_)) and by a slow carboxylation turnover rate (k*_cat_*^c^). In addition, Rubisco is a bifunctional enzyme that performs both carboxylase and oxygenase functions. The activity of oxygenase binds oxygen to RuBP instead of CO_2_, reducing the efficiency of carbon fixation. Among other things, some Rubisco enzymes have low CO_2_/O_2_ specificity (S_c/o_), indicating a low advantage of CO_2_ over O_2_ as a substrate. The oxygenation reaction can considerably lower photosynthesis efficiency and necessitates additional energy use for photorespiration. During photorespiration, the loss of CO_2_, which could be fixed in the RPP cycle, can reach 25–30% [1]. One ATP and one NADPH molecule is consumed to neutralize the products of photorespiration.

The aforementioned Rubisco properties are absolutely critical in a modern oxidative atmosphere with a low [CO_2_]/[O_2_] ratio. The low rate of diffusion of all types of C_i_ in water is another factor that makes the situation worse for aquatic species. For precisely these reasons, photoautotrophic cells have developed two adaptation strategies that allow them to sustain high photosynthetic productivity: (1) an increase in the affinity of Rubisco for CO_2_ with a simultaneous increase in the amount of this enzyme in the chloroplast; (2) an increase in the intracellular concentration of CO_2_ to compensate for the disadvantages of Rubisco characteristics—that is, the direct saturation of the enzyme with its substrate.

The first strategy is implemented in higher plants that fix carbon by the C_3_ type of photosynthesis. The second strategy is represented by three currently known ways of CO_2_ accumulation near Rubisco carboxylation centers. In higher land plants, it is achieved due to the schemes of C_4_- and CAM-photosynthesis. Aquatic photosynthetic microorganisms, namely, microalgae and cyanobacteria, perform C_3_-metabolism and operate another type of CO_2_ concentration, which is based on the direct “pumping” of C_i_ into a cell. All these strategies of CO_2_-concentration are known as “CO_2_-concentrating mechanisms” (CCMs). Here, we will use the term “CCM” to refer to the CO_2_-concentration mechanism in the cells of microalgae and cyanobacteria.

The abbreviation CCM is often deciphered as “carbon concentrating mechanism”, since its functioning solves two problems:Intracellular accumulation of C_i_ overcomes the problem of its low rate of diffusion on the way from the external environment to Rubisco. Thus, CCM solves the problem of the substrate limitation of photosynthesis;The concentration of CO_2_ molecules near the active sites of Rubisco compensates for its low substrate specificity.

CCM functions in cyanobacteria and eukaryotic algae assigned to different taxonomic groups [2,3,4]. The best studied CCMs belong to the model strains of cyanobacteria (such as *Synechocystis* sp. strain PCC 6803, *Synechococcus elongatus* PCC 7942, *Synechococcus* sp. strain PCC 7002) and green microalgae *Chlamydomonas reinhardtii* [5,6,7,8,9,10]. In addition to freshwater and marine species, CCM components are found in a number of alkaliphilic cyanobacteria [11,12]. It is noteworthy that these organisms grow in soda lakes with a very high content of bicarbonate, and, theoretically, they do not require any CO_2_ concentration. At the same time, alkaliphilic cyanobacteria are considered as direct relics of the ancient terrestrial microbiota that existed in the CO_2_-rich Archean atmosphere [13]. These circumstances raise the question about the time period and roots of the appearance of CCM on Earth.

Currently, there are several theories regarding the time of occurrence of CCM in cyanobacteria, ranging from ~350 Myr (million years) to ~3.5 Gyr (billion years) ago [2,14,15,16,17,18]. Experimental data indicate that the ancient form of CCM could function in these organisms even before the radical changes in the gas composition of the atmosphere ~2 Gyr ago [19]. Such a proto-CCM could compensate for the non-favorable [CO_2_]/[O_2_] ratio that occurs for various reasons in the pericellular layer [2,11,15,17]. It is possible that the absence of the early evolution of Rubisco [20,21] can be attributed to the presence of proto-CCM in ancient cyanobacteria.

In microalgae, the appearance of CCM is, most likely, independent. This is evidenced by the differences in the sets of CCM components of microalgae and cyanobacteria, as well as by the absence of homology between them. It was suggested that this event could have occurred about 400–500 Myr ago, after the appearance of photorespiration [22]. The development of alternate CO_2_-concentration strategies (C_4_- and CAM-photosynthesis) in terrestrial plants 20–30 Myr ago [23,24] is thought to have resulted from the inability to employ a “biophysical pump” to get C_i_ into the cells from the surrounding air.

The high ecological significance of CCM underpins the interest in it. Almost half of the oxygen produced on our planet is provided by oceanic phytoplankton, which includes cyanobacteria and microalgae, while the other half is provided by terrestrial plants [25,26]. The effectiveness of CO_2_ fixation is distributed similarly (1:1) between oceanic and terrestrial photosynthetic species in terms of their gross yearly primary output [25]. This fact becomes especially important in the present period of climate change, where mankind faces tasks of preserving the ecology of the biosphere, preventing the greenhouse effect and removing “excessive” amounts of CO_2_ from the atmosphere. The idea of harnessing the genetic potential of cyanobacteria and microalgae to introduce an artificial CCM into C_3_ plants in order to increase crop yields also dictates the relevance of versatile CCM research [10,27,28,29,30].

## 2. Rubisco, Types of Photosynthetic Carbon Metabolism and CO_2_-Concentrating Mechanisms

The interaction of organic and inorganic carbon cycles in autotrophic organisms occurs through photosynthetic C_i_ fixation in the RPP cycle with Rubisco as the key enzyme. The structure of Rubisco, as well as the phylogeny of these proteins, are fairly well investigated [21,31,32,33]. All Rubisco are divided into four basic forms (I–IV), each of which catalyzes the identical reaction, but has very different structural and kinetic characteristics [31]. An exception is Rubisco IV, which lacks catalytic activity. To distinguish the latter from active enzymes, proteins that belong to Rubisco IV are referred to as Rubisco-like-proteins. It is assumed that form IV is the progenitor of the CO_2_-fixing Rubisco [34].

Cells of higher plants, cyanobacteria and eukaryotic algae (along with proteobacteria) carry Rubisco form I [31]. Proteins of this group consist of large (L) (~55 kDa) and small (S) (~15 kDa) subunits, which form the L_8_S_8_ superstructure. The large subunit contains the enzyme’s active center, while the small subunit appears to provide structural stability and catalytic efficiency [32]. Rubisco forms I are subdivided into four groups (IA-ID) comprising enzymes with a different primary sequence of the large subunit. Of these, two forms, IA and IB, are present in proteobacteria, cyanobacteria, green algae and higher plants, while IC and ID are present in non-green algae and proteobacteria [35].

According to the mode of CO_2_ fixation, the following types of photosynthetic carbon metabolism are distinguished: C_3_, C_4_ and CAM. Only C_3_ higher plants do not require RPP cycle additions to maintain high photosynthetic productivity. In these organisms, Rubisco has a rather low turnover rate (k*_cat_*^c^), performing, on average, about three carboxylation reactions per second [21]. At the same time, these enzymes are characterized by a high affinity for CO_2_ as a substrate (avg. *K*_M_ (CO_2_)~14 μM) and by high CO_2_/O_2_ specificity (avg. S_c/o_~98). Furthermore, the amount of Rubisco in the chloroplast stroma in C_3_ higher plants is up to three orders of magnitude higher than that of CO_2_ and approximately equal to that of RuBP, which provides a high concentration of the enzyme’s active sites in vivo, reaching 10 mM [36,37]. These particular characteristics allow C_3_ higher plants to saturate the carboxylation reaction with CO_2_.

In contrast to C_3_ higher plants, cyanobacteria and microalgae cannot saturate Rubisco active centers at the current atmospheric CO_2_ concentration. Cyanobacterial Rubisco has a high carboxylation turnover rate (k*_cat_*^c^ up to 14 s^−1^), but a very low affinity for CO_2_ as a substrate (*K*_M_ (CO_2_) over 300 μM) and a low CO_2_/O_2_ specificity (avr. S_c/o_~48), indicating that CO_2_ has little advantage over O_2_ as a substrate [21,38]. The Rubisco of green microalgae are characterized by mean S_c/o_~62, with a higher affinity for CO_2_ than in cyanobacteria (avr. *K*_M_ (CO_2_)~32 μM), but a lower carboxylation turnover rate (avr. k*_cat_*^c^~3 s^−1^) [21]. In microalgae cells, Rubisco accounts on average for about 5% of the total cellular protein. The content of cyanobacterial Rubisco rarely exceeds 10%, which is five times less than in the photosynthetic tissues of C_3_ higher plants [39,40,41].

The Rubisco of CAM and C_4_ plants generally have kinetic parameters similar to those of C_3_ plants [21]. However, due to the peculiarities of their habitat and unique life strategies, these species exhibit significant diurnal changes in CO_2_ levels (CAM), or excessive photorespiration (C_4_) [23].

Thus, all photoautotrophs, other than C_3_ higher plants, require the adaptive mechanisms to maintain the efficiency of the RuBP carboxylation reaction. Those organisms use a common, most obvious strategy—the concentration of CO_2_ molecules near the active centers of Rubisco is used to saturate the enzyme. This strategy can be implemented in three different ways, which are discussed further below.

In higher terrestrial plants, CO_2_ concentration is achieved through the operation of C_4_- and CAM-photosynthesis, which differs from C_3_-photosynthesis in that they have the metabolic add-ons to the RPP cycle that complicate the biochemical pathways of CO_2_ fixation and reduction. This is why this type of CO_2_ fixation is called “biochemical” [3,18]. Certain species of freshwater aquatic plants also belong to CAM plants [23]. The processes of CO_2_ assimilation in C_4_- and CAM-photosynthesis are similar in many respects. In both C_4_- and CAM-plants, the first CO_2_ acceptor is phosphoenolpyruvate (PEP), which binds to the bicarbonate produced by the hydration of the CO_2_ molecule entering the cell (Figure 1). The primary products of photosynthesis here are C_4_-dicarboxylic acids, and the formation of CO_2_ for its incorporation into the RPP cycle (with a simultaneous concentration near Rubisco) results from subsequent decarboxylation reactions. Due to C_4_-photosynthesis, the concentration of CO_2_ at Rubisco localization sites is increased approximately 10–20-fold compared to the atmospheric level [42,43] resulting in the suppression of photorespiration. However, the contribution of C_4_ plants to the gross Earth photosynthesis is small, as they account for only 3% of all known species [44]. In terms of biochemistry and the general scheme, CAM photosynthesis is very similar to C_4_ photosynthesis, but it is slightly inferior in terms of the net CO_2_ fixation [45].

In cells of microalgae and cyanobacteria with a C_3_-type of photosynthesis, the CO_2_ concentration is achieved by the CCM, which ensures the active “pumping” of exogenous C_i_ into cells, with the subsequent accumulation of CO_2_ molecules at carboxylation sites. CCM was discovered in the 1980s [46,47,48,49,50,51,52]. Its scheme is referred to as a “biophysical” type of concentration [3,18,29,53]. In contrast to the constitutive mechanisms of CO_2_ concentration that function in C_4_- and CAM-plants, CCM is an inducible process that is activated only when the C_i_ concentration in the environment decreases. CCM is also an add-on to the RPP cycle with the exception that the C_i_ pool inside the cell does not enter into biochemical transformations before the RPP cycle, and the first CO_2_ acceptor is RuBP.

Characteristics of cyanobacterial and microalgal cells adapted to the atmospheric CO_2_ concentration are similar to that of C_4_ plants in terms of some photosynthetic parameters (reduced inhibition of photosynthesis by O_2_, reduced photorespiration and reduced CO_2_ compensation point). Cells grown at high CO_2_, however, show photosynthetic characteristics similar to those of C_3_ plants. CCM allows more efficient CO_2_ fixation in the RPP cycle [30] as compared to C_4_- and CAM-plants. Therefore, microalgae and cyanobacteria are the essential contributors to the gross photosynthesis, organic matter and oxygen formation in our planet [25,26].

It should be emphasized that the literature currently refers to C_3_ higher plants as having the so-called “passive CO_2_-concentrating mechanism” (pCCM) [54]. This term refers to the ATP-independent biological transport processes that ensure the capture of CO_2_ released during respiration and photorespiration, its subsequent delivery to the Rubisco and re-fixation. Moreover, it has been proposed that cells of C_3_ higher plants may employ a “basal CCM” depending on the functioning of mitochondrial CAs [55]. This scheme also implies the prevention of CO_2_ leakage from a cell and allows the efficient recycling of CO_2_ released due to the reactions of the tricarboxylic acid (TCA) cycle and photorespiration. However, the two above-mentioned schemes cannot be considered as a true CO_2_-concentration, which implies the energy-dependent transformation of the low concentration of exogenous CO_2_ into its high intracellular concentration near the active centers of Rubisco. Instead, the so-called “basal” and “passive” CCM are linked to resource conservation and its reasonable use in cells.

## 3. General Principles of CCM Operation

The CCM is based on the co-operation of the C_i_ (HCO_3_^−^/CO_2_) uptake systems and carbonic anhydrases (CAs). A simplified scheme of the CCM operation is shown in Figure 2. The first step is the creation of a HCO_3_^−^ pool, the intracellular content of which may be several orders of magnitude higher than the concentration of C_i_ in the environment. In the second step, the stored HCO_3_^−^ is used to generate a high concentration of CO_2_ molecules near the active centers of Rubisco.

In general, the CCM of cyanobacteria and microalgae is characterized by the following structural and physiological features:The CCM is organized within a single cell;The CCM is an inducible process: it is activated when the concentration of exogenous C_i_ is insufficient to ensure the efficient photosynthesis;The CCM induction requires light;In contrast to the C_4_- and CAM-photosynthesis, and the C_3_-photosynthesis of higher plants, where C_i_ enters the cell only by CO_2_ diffusion, microalgae and cyanobacteria additionally possess the systems of the active (energy-consuming) uptake of both CO_2_ molecules and HCO_3_^−^ ions;During the operation of the CCM, the rapid interconversion of C_i_ species (CO_2_/HCO_3_^−^) both outside and inside the cell is maintained by CAs;The efficient incorporation of CO_2_ into the RPP cycle is achieved through the joint localization of Rubisco and CA in special microcompartments, such as carboxysomes in cyanobacteria or pyrenoid in microalgae. The cooperation between Rubisco and CA is the fundamental basis for the functioning of the CCM;The CCM includes the prevention of CO_2_ leakage from the cell;The structural features of the CCM allow one to protect Rubisco from O_2_ and to minimize the oxygenase reaction.

The CCM in cyanobacteria and microalgae is known as the “biophysical” CO_2_ concentration because it is primarily based on the activation of systems for an active physical uptake of exogenous C_i_. At the same time, cells under conditions of C_i_ starvation undergo rather complex rearrangements in structural organization and biochemical processes [6,22,56,57,58,59,60,61].

It should be emphasized that CCM is a strictly energy-dependent and light-regulated process, since the absorption and accumulation of C_i_ by the cell does not occur in the dark [50]. In addition to intracellular C_i_ accumulation, light regulates many other processes related to CCM functioning—such as the transcription of CCM-associated genes, relocalization of the CCM components with the cell, formation of carboxysomes and pyrenoids, etc. [6,7,22,58,61,62,63]. It is currently unknown how the light regulation of the CCM induction and operation is performed. It has been suggested that the activity of C_i_ uptake systems can be regulated allosterically depending on changes in the [ATP]/[ATP + ADP + AMP] pools’ ratio under transitions from the light to darkness [63].

The energy expenditure of the cell for the CCM requires special consideration. C_i_-limiting conditions cause a significant decrease in the growth rate, which is most likely due to the CCM’s high energy requirements, which are primarily related to the operation of the induced C_i_ transporters [6]. At the same time, the CCM is generally an energetically advantageous mode of photosynthetic assimilation. Indeed, by inhibiting photorespiration, CCM helps to save energy: the energy cost for the CCM functioning is lower than for the maintenance of photorespiratory metabolism [64]. The CCM also reduces (compared to C_3_ plants) the amount of Rubisco per unit biomass required to ensure a given rate of CO_2_ fixation. As a result, the amount of cell resources used for Rubisco synthesis is similarly decreased [64].

In addition, the functioning of the CCM helps to maintain an equilibrium between light and dark photosynthetic reactions. With the onset of C_i_ starvation, the activity of RPP cycle decreases, which leads to a decrease in the available NADP^+^ and ADP molecules. A lack of NADP^+^, which is required to keep the electron transport chain running, leads to the suppression of light processes by the inhibition of the excessive activity of the photosynthetic apparatus (photoinhibition). The suppression of the electron transport can also result from excessive proton accumulation in the thylakoid lumen due to a shortage of ADP molecules and a decrease in ATP synthase activity. The activation of CCM leads to the restoration of RPP cycle activity. Simultaneously, the conversion of bicarbonate to CO_2_ for Rubisco using lumen protons minimizes their excessive accumulation [60,65,66,67]. Thus, CCM can improve the efficiency of light utilization and, as a result, the overall efficiency of the photosynthetic machinery [64].

### 3.1. Exogenous Sources of C_i_, C_i_ Uptake Systems and the Formation of the Intracellular HCO_3_^−^ Pool

Microalgae and cyanobacteria live in an aquatic environment where C_i_ exists in three different forms: CO_2_, HCO_3_^−^ or CO_3_^2−^, whose ratio depends on the pH of a solution [68] (Figure 3). Seawater with alkaline pH values contains a high pool of bicarbonate ions (1.8 mM, 90%) along with 10% (0.35 mM) of CO_3_^2–^ and a small amount (0.01–0.02 mM) of dissolved CO_2_ in the equilibrium with air [49,69]. The C_i_ total concentration in seawater can reach 2 mM. Freshwater contains only 10 μM C_i_, and its available forms are represented by CO_2_ and HCO_3_^−^. As an external source of C_i_, microalgae and cyanobacteria use CO_2_ and/or HCO_3_^−^. Many microalgae and cyanobacteria exhibit species-specific preferences for some forms of C_i_ [70]. The preferential form of exogenous C_i_, as well as the set of C_i_ uptake systems specific to a particular cell, determine the preference for CO_2_ or HCO_3_^−^ uptake.

The outer membrane is the first barrier for exogenous C_i_, which can pass it through aquaporin channels [2,5]. C_i_ must then cross the plasma membrane and, in eukaryotic algae, the chloroplast membranes. Due to its high solubility in lipids, the CO_2_ molecule can enter the cell by direct diffusion. In contrast to CO_2_, negatively charged HCO_3_^−^ can only cross the cell membranes via active transport. At the same time, even in the presence of a concentration gradient, HCO_3_^−^ is well retained in a cell. The presence of bicarbonate uptake systems in microalgae and cyanobacteria is critical, because these organisms live in aquatic environments where CO_2_ diffusion rates are much lower than in air, and HCO_3_^−^ often becomes the predominant form of exogenous C_i_ [4].

In general, three fluxes can cause the intracellular accumulation of C_i_ (Figure 2):Active transport of HCO_3_^−^ into a cell;Entry of CO_2_ into a cell by the diffusion of dissolved gas;Increased CO_2_ diffusion into a cell as a result of the quick conversion of CO_2_ (which has already entered the cell) into HCO_3_^−^.

Operation of the C_i_ uptake system results in the intracellular accumulation of C_i_ as a pool of HCO_3_^−^. The conversion of CO_2_ to HCO_3_^−^ in intracellular compartments is dictated by their alkaline pH values and by the catalytic action of C_i_ conversion systems based on the activity of CAs (Section 3.2). In general, the intracellular C_i_ concentration accumulated due to CCM can be 10–1000 times higher than that in the environment [49,71].

The active cellular uptake of C_i_ against the concentration gradient is an energy-dependent process [5,64]. This uptake is supported by ATP molecules, which are generated during photosynthetic electron transport, both cyclic and linear. When cells adapt to low CO_2_ concentrations, the cyclic electron transport activity increases, allowing an increase in the ratio of ATP/NADPH produced by the electron transport chain and providing an additional ATP influx for CCM [35,72]. An increase in the ratio of photosystem activities (PSI/PSII) in microalgae has been demonstrated under C_i_-limiting conditions [73,74]. In cyanobacteria, the activity of NADPH dehydrogenase complexes, located in the thylakoid membrane and participating in the cyclic electron transport around PSI, also increased [35].

### 3.2. Carbonic Anhydrases and CO_2_/HCO_3_^−^ Conversion Systems as an Element of the CCM

The carbonic anhydrase (CA, EC 4.2.1.1) system, consisting of external and intracellular forms of the enzyme, is the second important element of the CCM of microalgae and cyanobacteria. The reaction catalyzed by this enzyme determines the CA participation in biological processes: CO_2_ + H_2_O ⇆ HCO_3_^−^ + H^+^. Such an interconversion is possible without a CA, but it is extremely slow. For example, the value of the rate constant for the spontaneous reaction of CO_2_ hydration is 0.037 s^−1^ [75], whereas for the reverse reaction of the HCO_3_^−^ dehydration it is 20 s^−1^ [76]. Such a circumstance could significantly slow down biochemical processes. This is why CA is such an important enzyme of carbon metabolism with a fundamental significance of carbon-based life [77]. CAs are found in all presently known systematic groups of living organisms; they are present in all organs, tissues and compartments, where the acceleration of CO_2_/HCO_3_^−^ interconversions or a rapid change in concentration of any of the four reaction components is required. The detailed information regarding the mechanism of catalysis, diversity of structural organization, phylogeny and biological functions of CAs in living organisms is properly reviewed in several books and articles [78,79,80,81].

Depending on the primary protein sequence, its three-dimensional structure, active site organization and catalytic properties, all known CAs are divided into eight classes—α, β, γ, δ, ζ, η, θ and ι [79]. Cyanobacteria and microalgae possess three classes of CAs—α, β and γ. In addition to these, microalgae cells contain CAs of δ, ζ and θ-classes [82]. CAs of the same class may be represented by several proteins with different biological relevance. The most prominent physiological roles of CAs in cyanobacteria and microalgae are related to the maintenance of CCM and photosynthesis in their cells. The enzymes take part or play the key roles in the following processes associated with these global mechanisms:C_i_ uptake by supplying CO_2_/HCO_3_^−^ molecules for their transport across cell membranes;Photosynthetic C_i_ fixation by generating the CO_2_-substrate for Rubisco;Prevention of CO_2_ leakage from the cell by the operation of intracellular C_i_ conversion systems possessing CA activity;Regulation of photosynthesis by the modulation of the PSII activity and electron flow rate via the protection of the microalgal water oxidizing complex (WOC) from excess protons.

The fundamental basis for the efficient operation of CCM is the rapid compartment-specific transformation of the C_i_ species with the participation of CAs. Equilibrium concentrations of CO_2_ and HCO_3_^−^ in the enzymatic reaction correlates with the distribution of C_i_ forms in the solution depending on its pH, as described by the Henderson–Hasselbach equation: pH = 6.3 + lg([HCO_3_^−^]/[CO_2_]) [68]. Thus, at pH < 6.3, the equilibrium shifts toward CO_2_ formation, whereas at pH > 6.3, the equilibrium shifts toward the predominant generation of HCO_3_^−^ (Figure 3). Similarly, the predominance of a specific C_i_ form in a particular cellular compartment depends on its pH. The cytoplasmic pH in different species of microalgae and cyanobacteria ranges from 7.1 to 8.2 [5,83,84,85]; the pH of chloroplast stroma is about 8 [85]. In these compartments, C_i_ is mainly represented by bicarbonate (about 80–90%). Whereas, in lumen, when a proton gradient is created in the illuminated thylakoid membranes, the pH reaches 4–5 units [85], and CO_2_ becomes the main form of C_i_. Mathematical modeling shows that the cyanobacterial carboxysome matrix also possesses an acidic pH [86], which can be formed as a result of proton release during the RuBP carboxylation reaction [87,88,89]. Lower pH values of the internal contents of carboxysomes (in comparison to the cytoplasm) have recently been demonstrated [90].

Thus, the direction of C_i_ flow from the environment to Rubisco is insured by the difference in the physico-chemical properties of CO_2_ and HCO_3_^−^, as well as by the difference in pH in the cellular compartments and rapid transformation of C_i_ with the participation of CAs (Figure 2). The involvement of CAs in the CO_2_/HCO_3_^−^ delivery for their uptake systems is mediated by the local pH values of the periplasmic or pericellular space and by the advantage of a hydration or dehydration direction in the enzymatic reaction. The weakly alkaline pH of the cytoplasm and chloroplast stroma promotes the preferential accumulation of C_i_ as a lipophilic-neutral pool of HCO_3_^−^. The latter significantly reduces the risk of the spontaneous leakage of C_i_ from a cell. Conversely, under the acidic pH of the carboxysomes matrix or the lumen of the intrapyrenoid thylakoids, CAs “release” CO_2_ molecules from the HCO_3_^−^ pool for their use by Rubisco [85,86,91,92]. The benefit of the bicarbonate dehydration reaction under acidic lumen pH values enables the enzyme’s protective activity in H^+^-removing from WOC in stromal thylakoids [92,93,94].

The operation of intracellular C_i_ conversion systems, presumably based on their CA activity, is critical for preventing CO_2_ molecule leakage into the external environment (Section 4.2 and Section 5.3.4). These systems, located in the cytoplasm in cyanobacteria and in the stroma of chloroplasts in microalgae, convert CO_2_ molecules into HCO_3_^−^, promoting the “locking” of C_i_ inside the cell. Concurrently, the C_i_ conversion systems capture CO_2_ molecules that have not been incorporated into the RPP cycle.

### 3.3. Rubisco-Containing Microcompartments: Carboxysomes and Pyrenoids. The Principles of Cooperation between CA and Rubisco

All of the stages of CCM described above (exogenous C_i_ uptake with the participation of external CAs, and its accumulation as HCO_3_^−^ in the cytoplasm in cyanobacteria or in the cytoplasm and stroma of chloroplast in microalgae) are prerequisites for the final stage: the conversion of the accumulated pool of HCO_3_^−^ into CO_2_ molecules that serve as Rubisco substrate. To effectively engage CO_2_ in the RPP cycle and minimize the leakage of these molecules from a cell, CO_2_ concentration increases only locally, near Rubisco active centers (Figure 2). The intracellular CAs are responsible for the conversion of HCO_3_^−^ to CO_2_. A microcompartment, where the majority of Rubisco is concentrated and where this enzyme is co-localized with CA, is a structural feature of CCM [35,74,95,96]. In cyanobacteria, these microcompartments are known as carboxysomes, while in microalgae, they are known as pyrenoids. The Rubisco content in the inner phase of pyrenoids and carboxysomes reaches 70–98% of its total amount in the cell [57,97]. Thus, Rubisco and CA form the central element of the CCM, “for whose benefit” this mechanism functions. The interplay of the organic and inorganic carbon cycles in microalgae and cyanobacteria is mostly mediated by this tandem. A CCM model, in which CA and Rubisco were co-localized in carboxysomes, was first proposed in 1989 [98].

Depending on the C_i_ availability, both carboxysomes and pyrenoids undergo significant changes in size, composition and structure. The pyrenoid structure becomes more apparent under low carbon conditions, and it is typically surrounded by a thick starch sheath [58,85,99]. Under these conditions, the majority of the entire amount of cellular Rubisco is present in the pyrenoid [61,99]. At a high CO_2_, however, less than 50% of the enzyme occurred in the pyrenoid, and the remaining percentage is found in the chloroplast stroma together with the accumulated starch grains. In cyanobacteria, C_i_-limiting conditions lead to an increase in the number of carboxysomes; they become more contrasting due to the synthesis of a protein shell [57,100]. Rubisco is mostly concentrated in the carboxysomes during C_i_ deficiency, much like a pyrenoid.

New pyrenoids are formed by both a fission and de novo assembly [101]. In contrast to carboxysomes, pyrenoids are a transient subcellular feature. The CCM is present in all pyrenoid-containing microalgae, but not all microalgae that accumulate C_i_ under low carbon conditions have pyrenoids [2]. As a result, the subtleties of pyrenoid involvement in CO_2_ concentration have yet to be evaluated. The identification of the factors that regulate the pyrenoid assembly and functioning is currently an active area of study [102,103,104,105,106].

## 4. CCM of Cyanobacteria

Photosynthetic gram-negative bacteria, namely, cyanobacteria, are among the oldest organisms on our planet, and the only prokaryotes capable of oxygenic photosynthesis. Due to the CCM, cyanobacteria have extremely high photosynthetic productivity. According to various estimates, these organisms contribute 10–30% of the global net primary carbon fixation [107,108,109].

The current knowledge of cyanobacterial CCM is based on studies of model laboratory strains such as the freshwater *Synechocystis* sp. strain PCC 6803 and *Synechococcus elongatus* PCC 7942, or the marine *Synechococcus* spp. and *Prochlorococcus* spp. Principles of the CCM function have been outlined in a number of previous reviews [5,6,7,17,110]. Here, we will discuss only the recent progress in the field and review the general scheme of the induction and functioning of the cyanobacterial CCM, taking into account the principles of operation of its individual components.

### 4.1. CCM Induction in Cyanobacteria

As previously stated, the cyanobacterial CCM is an inducible mechanism, which is solely activated when the exogenous C_i_ is insufficient to saturate the dark phase of photosynthesis. In cyanobacteria, allophycocyanin may play the role of the C_i_ sensor [111]. Alternatively, this role is assigned to the soluble adenylate cyclase, whose activity is directly proportional to the concentration of C_i_ in the environment [112,113,114]. The cAMP synthesized by adenylate cyclase can work as a C_i_-sensing signal.

Two states, “low-affinity” (basal) and “high-affinity” (induced), are characteristic of the cyanobacterial CCM [5,6]. The basal level of CCM is kept in cells that grow at the so-called “high” C_i_ level, and is capable of maintaining the efficiency of the RuBP carboxylation reaction in the RPP cycle. In the laboratory, these conditions correspond to the growth of cultures at 1.5–2% CO_2_ in a gas–air mixture. Here, the CCM operation is ensured by its constitutive components, which include low-affinity high-rate C_i_ uptake systems (NDH-1_4_ and BicA), Rubisco and carboxysomal CA. The accumulation of intracellular C_i_ and concentration of CO_2_ molecules near Rubisco do not occur under these conditions.

The induced state of the CCM corresponds to the C_i_-limiting conditions. These usually refer to the growth in the most natural aquatic environments, including seawater. High-affinity C_i_ uptake systems (NDH-1_3_, SbtA and BCT1) are primarily responsible for the operation of the induced CCM [5], ensuring the accumulation of the intracellular HCO_3_^−^ pool. In addition, the increased level of the thylakoid β-CA, EcaB, is recorded under these conditions [115]. This occurrence is explained by the involvement of EcaB in the operation of the NDH-1_3_ complex.

The CCM induction entails for optimizing the set of its components and/or changing their activity. This is ensured by three levels of regulatory processes: (1) transcriptional, which regulates the mRNA level of the CCM components; (2) post-transcriptional, which regulates the translation efficiency from mRNA; and (3) post-translational, which tunes the activity of already existing protein components through their modifications or by allosteric regulation.

The most important biochemical changes in response to low-C_i_ stress have been analyzed in recent reviews [6,7,116]. Carbon deficiency inhibits the RPP cycle and causes an imbalance in photosynthesis’s light and dark reactions. Photodynamic disturbances lead to changes in the redox state of a cell. Protein synthesis becomes suppressed. Simultaneously, photorespiration is activated as a result of a shift of Rubisco activity towards oxygenation. Glycolysis, the TCA cycle, and the oxidative pentose phosphate cycle are activated, allowing the additional synthesis of NADPH and ATP, as well as the release of carbon from organic compounds for its utilization in de novo protein synthesis. Intermediates in these metabolic pathways can act as effectors, correcting the ability of the CCM-associated transcriptional factors (TFs) to bind to DNA and exert control over the gene expression [6,7]. Among these metabolites with established regulatory functions are RuBP, 2-phosphoglycolate (2-PG) and α-ketoglutarate (α-KG; also known as 2-oxoglutarate). Additionally, the regulation of gene expression can be achieved by altering the DNA topology [117,118].

The cyanobacterial cell acclimation to low C_i_ is controlled by three known TFs [6,7,116]: CmpR, NdhR (CcmR), and CyAbrB2. CmpR (*cmp* operon regulator) acts as a transcriptional activator of the *cmp* operon, encoding the high-affinity bicarbonate transporter, BCT1, of freshwater cyanobacteria [119]. CmpR binding to the promoter is enhanced in the presence of RuBP and 2-PG, whose level is expected to increase under low CO_2_ [120].

Another TF, NdhR (CcmR) (NDH-1 genes regulator/carbon concentrating mechanism regulator), acts as a repressor of genes, encoding: (a) C_i_ uptake systems SbtA, BicA and NDH-1_3_; (b) SbtB regulator protein; and (c) *mnh* operon for the complex involved in the generation of an electrochemical Na^+^ gradient required for the Na^+^-dependent HCO_3_^−^ uptake [121,122,123,124]. A molecule of α-KG acts as a co-repressor and enhances the promoter affinity of NdhR, whereas 2-PG is a CCM inducer [125]. It was originally reported that NADP^+^ could also act as a co-repressor of NdhR [126], but further studies have not confirmed this fact [125]. The concentration of α-KG is related to the carbon/nitrogen balance of the cell. These molecules are accumulated under the C_i_ excess due to the active TCA cycle and under a nitrogen limitation [116,127,128]. At the same time, the 2-PG molecules generated by the Rubisco oxygenation reaction serve as a signal of C_i_ deficiency. When 2-PG appears, it antagonizes the effect of α-KG, leading to the dissociation of NdhR from its target repressing site [125].

A third TF, cyAbrB2 (cyanobacterial AbrB-like protein 2), acts complementarily to CmpR and NdhR, through controlling several CCM-related genes encoding BCT1, NDH-1_3_ and SbtA/B [129,130]. The effectors that regulate this TF are currently unknown.

At the post-transcriptional level, the variation in the composition of CCM components is regulated translationally by small regulatory RNAs [131], which help to enhance or weaken the synthesis of certain proteins. Small RNAs may control the increase in the number of carboxysomes under C_i_ deficiency, since the expression of the *ccm* gene cluster was not up-regulated under these conditions [6].

At the post-translational level, the regulation may operate via changes in the thiol groups status due to changes in the redox state of a cell [7], phosphorylation [132] or other structural modifications of proteins. An important regulatory element is the SbtB protein, a member of the PII superfamily of multifunctional signal processing proteins [133,134]. Previously, SbtB was considered only as a regulator, which allosterically controls the transport activity of the HCO_3_^−^ transporter, SbtA [63,116,135,136]. The binding of SbtB to various adenyl-nucleotides, including cAMP—a potential signaling molecule involved in C_i_ acclimation [113,114], suggested its involvement in the modulation of the CCM activity [133]. The participation of SbtB in the regulation of the expression of a number of CCM-associated genes (regardless of the involvement of secondary messengers) has recently been reported [137]. The mechanism of this regulation remains to be evaluated.

### 4.2. C_i_ Uptake in Cyanobacteria and C_i_ Conversion System in the Cytoplasm

There are five systems known in cyanobacteria model strains that allow C_i_ entry into the cell [7,138]. These include three transporters of HCO_3_^−^ and two CO_2_ uptake systems. The latter ensure the diffusion of CO_2_ inside the cell due to its rapid conversion to HCO_3_^−^ and creation of a negative gradient (Figure 4). The auxiliary elements (Figure 4) supporting the C_i_ transport systems in cyanobacteria are: (1) the Na^+^/H^+^ antiporter NhaS3 and (2) the specialized NDH-1 complex, Mnh; both participate in the creation of a Na^+^ gradient during the Na^+^-dependent transport of bicarbonate; as well as (3) the proton pump PcxA, which works to release H^+^ from the cytoplasm and contributes to the maintenance of cellular pH at weakly alkaline values, thus ensuring the accumulation of the intracellular pool of C_i_ in the form of HCO_3_^−^.

Bicarbonate transporters are represented by the proteins SbtA, BicA and the BCT1 complex located in the plasma membrane [138]. Notably, cyanobacteria lack the DAB complex responsible for C_i_ uptake in many prokaryotes, including chemolithoautotrophs [139,140]. Energy equivalents for the active HCO_3_^−^ uptake are ATP molecules (required for BCT1 operation) or the electrochemical gradient of Na^+^ that enables the SbtA and BicA operation [7].

The high-affinity (*K*_M_ (HCO_3_^−^)~15 μM) transporter BCT1 (bicarbonate transporter 1) with a medium flux rate is encoded by the *cmpABCD* operon [141]. BCT1 is expressed only under conditions of strict C_i_ limitation [100,142,143]. In marine and alkaliphilic species inhabiting environments with high HCO_3_^−^ concentrations, this high-affinity transporter is detected quite rarely [5,12,144].

SbtA (sodium-bicarbonate transport A) is an inducible, high-affinity (*K*_M_ (HCO_3_^−^)~5 μM) Na^+^-dependent transporter with a relatively low flux rate [145]. Cyanobacterial genomes may contain several copies of the *sbtA* gene with varying degrees of homology. The SbtA transport activity is allosterically regulated by the PII-like signaling protein SbtB, which can bind a number of adenyl nucleotides [63,133]. Two corresponding genes, *sbtA* and *sbtB*, are often adjacent in a single operon and are expressed together under C_i_-limiting conditions [133]. The binding of SbtB to SbtA suppresses the bicarbonate uptake function of SbtA [135]. The formation and dissociation of the SbtA/SbtB complex depends on the nature of the SbtB-bound adenyl nucleotide. There are two different views on the nature of the regulation of SbtA activity by SbtB. One of them [116,136] suggests that SbtB acts as a C_i_ sensor distinguishing between [ADP + AMP] and [cAMP] concentration differences under low and high C_i_, respectively. The second model [63] implies that SbtB performs a light-dependent regulation of SbtA via changes in the adenylate charge ratio ([ATP]/[ATP + ADP + AMP]) in a cell under its transition from light to darkness.

A third HCO_3_^−^ transporter, BicA (bicarbonate transporter A), was characterized by low-affinity (*K*_M_ (HCO_3_^−^) 90–170 μM) and supported a high flux rate [146]. In different species of cyanobacteria, BicA can be expressed constitutively or be highly inducible under a C_i_ limitation [100,123,146,147]. Often, similarly to SbtA, cells contain several BicA proteins with varying homology. Remarkably, BicA is the only C_i_ transport system usually detected in marine cyanobacteria [5].

Two systems, NDH-1_3_ (NDH1-MS) and NDH-1_4_ (NDH1-MS′), are the special modification of NADPH dehydrogenase respiratory complexes (NDH-1) that occur in eubacteria and eukaryotic mitochondria [148]. NDH-1_3/4_ is responsible for CO_2_ uptake in cyanobacteria [149,150]. In contrast to the mitochondrial complexes, which generate a H^+^ gradient for ATP synthesis simultaneously with NADPH oxidation, NDH-1_3/4_ use the reduced ferredoxin as the energy equivalent. In this regard, the designation for cyanobacterial “NDH-1” has recently been changed to “photosynthetic complex I” [151].

In microalgae, both complexes NDH-1_3_ and NDH-1_4_ are localized in thylakoid membranes [152,153]. Low-affinity (*K*_M_ (CO_2_)~10 μM) NDH-1_4_ is expressed constitutively, whereas high-affinity (*K*_M_ (CO_2_)~1–2 μM) NDH-1_3_ is induced by the C_i_ limitation [5]. It is thought that NDH-1_3/4_ converts CO_2_ entering the cells into HCO_3_^−^ due to the CA activity of their unique (in comparison with other NDH-1 complexes in cyanobacteria) constituent proteins, CupA/B, which are also known as ChpY/X [149,150,154]. Although the CA specific activity of CupA/B has not been confirmed experimentally, this hypothesis was well supported by the computer simulations as well as by a number of indirect data [154,155,156,157]. Two alternative hypotheses have been proposed regarding the functioning of NDH-1_3/4_ complexes. Their fundamental difference lies in the assumption of whether the CO_2_ substrate is supplied to the CupA/B active centers: (a) from the cytoplasm [115] or (b) from the thylakoid lumen [158]. It has also been suggested that the CA activity of NDH-1_3/4_ may be maintained or regulated by β-CA and EcaB, whose specific activity and interaction with CupA/B proteins have been demonstrated [115].

In addition to the CO_2_ uptake, NDH-1_3/4_ complexes act as C_i_ conversion systems and prevent the leakage of CO_2_ molecules that escape from carboxysomes (Figure 4). Under high-light stress, NDH-1_3/4_ can participate in the dissipation of excess HCO_3_^−^ and light energy by relieving the plastoquinone pool over-reduction and preventing photoinhibition [115]. This phenomenon of C_i_ backflow into the environment to dissipate the excess light energy by the rapid recycling of ATP molecules or their equivalent (proton gradient) was first described about 25 years ago [159,160] and designated as “C_i_ cycling”. Moreover, NDH-1_3/4_ complexes participate in cycling the electron flow around PSI [156,161].

### 4.3. Cyanobacterial Carbonic Anhydrases

Cyanobacterial cells contain external (located in the outer layers relative to the plasma membrane), thylakoid and carboxysomal forms of CAs. However, not all of them are involved in the CCM.

The EcaA (external carbonic anhydrase, alpha class) protein, which possesses a leader sequence for transport through the plasma membrane, is the only CA of α-class in cyanobacteria. The *ecaA* gene is found in the genomes of many cyanobacteria, but the presence of EcaA in cells has been confirmed only in two species, *Anabaena* sp. PCC 7120 [162] and *Cyanothece* sp. ATCC 51142 [147]. Earlier studies also reported the presence of EcaA in *Synechococcus elongatus* PCC 7942 [162]. This conclusion was based on the Western blot analysis of the *Synechococcus* total proteins with antibodies against EcaA from *Anabaena*. However, the specific antibodies could not locate EcaA in *Synechococcus* cells, regardless of the C_i_ supply [163].

The EcaB protein (external carbonic anhydrase, beta class) was also previously referred to as the external CA, because of the presence of a hypothetical lipoprotein lipid attachment site [164] and a leader sequence for transport through the plasma membrane [165]. Although the presence of EcaB in the periplasmic space of *Synechocystis* sp. PCC 6803 was previously directly confirmed by proteomics’ methods [165], a recent study demonstrated that the bulk of this protein is associated with thylakoid membranes, and only a small part is associated with the plasma membrane [115].

The specific activities of EcaA and EcaB have only recently been confirmed [115,147,163], more than 20 years after the discovery of these proteins. The physiological role of EcaA, as well as that of periplasmic EcaB, has not yet been defined. The biological role of the thylakoid form of EcaB is probably related to the function of NDH-1_3/4_ systems responsible for CO_2_ uptake and C_i_ conversion in the cytoplasm [115].

Another external β-CA mentioned in the literature is the protein CahB1 (carbonic anhydrase, beta class, protein 1) from the haloalkaliphilic cyanobacterium *Sodalinema gerasimenkoae* IPPAS B-353 (formerly, *Microcoleus chthonoplastes*), which appeared in the cell envelopes and exopolysaccharide layer [166,167]. While CahB1 has an extremely high degree of homology with the carboxysomal β-CA, CcaA, the presence of CahB1 in *S. gerasimenkoae* carboxysomes was not confirmed. The paradox was reinforced by the fact that CahB1 does not contain any classical leader peptide for transport through the plasma membrane. Moreover, it appears that CahB1 may be the only active CA in the IPPAS B-353 strain [144]. If so, the question about the CA that delivers CO_2_ to Rubisco in *S. gerasimenkoae* carboxysomes remains open. Notably, when expressed heterologously in *Synechocystis* sp. PCC 6803, CahB1 acts as a functional carboxysomal CA [168]. Theoretically, the overall data [144,166,168] could be explained by the potential variations in the architecture of the CAs’ system in cyanobacteria from different habitats. The “non-model” strains, including *S. gerasimenkoae*, may exhibit different schemes of CA involvement in the photosynthetic carbon metabolism compared to “model” strains. At the same time, *S. gerasimenkoae* belongs to a group of the so-called “relict” cyanobacteria [11,13]. Therefore, there is a possibility that those cells preserved a variant of the ancient *proto-secretion* system [168].

The carboxysomal CAs of cyanobacteria are represented by the following enzymes: β-CA CcaA (carboxysomal carbonic anhydrase protein A), formerly IcfA (inorganic carbon fixation protein A); β-CA CsoSCA (carboxysome shell carbonic anhydrase), formerly CsoS3 (carboxysome shell protein 3); and γ-class CA, CcmM (carbon concentrating mechanism, protein M) [169,170,171,172]. The role of these CAs is to convert HCO_3_^−^ to CO_2_ for its further fixation by Rubisco and involvement in the RPP cycle.

### 4.4. Carboxysomes

The Rubisco-containing microcompartments in cyanobacteria are called carboxysomes. Carboxysomes are polyhedral protein microbodies, usually of 100–400 nm in diameter, located in the cytoplasm and covered by a single-layer protein shell [173,174,175,176,177,178]. On the transmission electron microscopy images, carboxysomes appear as polyhedral electron-dense inclusions in the cytoplasm (Figure 5a). The presence of carboxysomes is also a characteristic feature in many autotrophic bacteria [177].

Carboxysomes may contain two Rubisco subclasses, IA and IB, according to which they are called α- or β-carboxysomes, and the organisms containing them are called α- and β-cyanobacteria, accordingly [174,175]. These two types of carboxysomes are structural and functional analogues, and are composed of homologous proteins and enzymes. In addition, carboxysomes of α- and β-type differ in their organization at the genome level. The genes encoding polypeptides included in α-carboxysomes are usually organized in a single *cso* operon, whereas the genes for the components of β-type carboxysomes are clustered in several genomic locations [174,175]. The clustering of β-carboxysome genes implies their independent regulation and plasticity of expression in response to environmental changes [177].

Recent studies suggest the independent origin of α- and β-carboxysomes as a result of convergent evolution [179]. Little is known about the potential ecological benefits of each type of these microbodies. Until recently, it was thought that cyanobacteria with the Rubisco form IA mostly inhabit sea waters, whereas most cyanobacteria with the Rubisco form IB prefer fresh water environments. However, recent studies challenge this paradigm [180].

Both types of carboxysomes consist of a matrix and a protein shell (Figure 5b). The matrix includes Rubisco and structuring proteins that support the three-dimensional organization of the enzyme (CsoS2 or CcmM in α- and β-carboxysomes, respectively). Structural units of the carboxysome’s shell are several types of homo-oligomeric proteins that form shell vertices or shell facets with pores, through which metabolite molecules can pass. It is suggested that the pores facilitate the selective entry of HCO_3_^−^ and RuBP inside this microcompartment and the simultaneous release of 3-phosphoglyceric acid (3-PGA) into the cytoplasm [181]. Molecular simulations indicate that the carboxysome shell functions as a barrier for CO_2_ and O_2_ [182,183]. It blocks CO_2_ leakage from the matrix and, at the same time, prevents O_2_ entry, thus keeping carboxylase activity of Rubisco at the high level.

The carboxysomal β-CAs, CsoSCA and CcaA are functional analogues. CsoSCA is bound to the inner side of the α-carboxysomal shell, and probably carries out the conversion of HCO_3_^−^ to CO_2_ independently from other protein components [171]. Unlike CsoSCA, CcaA functions as part of the so-called “bicarbonate dehydration complex” located on the inner layer of the β-carboxysomal shell [184]. Another important protein of β-carboxysomes is CcmM. CcmM is expressed in two forms [185]. The full-length form consists of an *N*-terminal γ-CA domain followed by several RbcS-like domains attached by linkers. The short form of CcmM contains only RbcS-like domains. The full-length CcmM is part of the bicarbonate dehydration complex, and it can also function as CA in the absence of the *ccaA* gene [186,187]. The short form of CcmM serves as the structural element responsible for the paracrystalline organization of Rubisco IB [174,188]. Recently, along with the short form, the presence of full-length CcmM was shown in the carboxysome matrix, which can probably also participate in the three-dimensional organization of Rubisco [189].

Another component of cyanobacterial carboxysomes is Rubisco activase. In α-carboxysomes, this component is represented by the CbbX protein [190]. Cells of β-cyanobacteria can contain ALC (activase-like cyanobacterial protein), which does not function as a canonical Rubisco activase, but its presence is important for the proper assembly of carboxysomes [191].

The impairments in carboxysome distribution in cells cause their elongation, asymmetric division, increased Rubisco levels and growth retardation [192]. These observations suggest that carboxysomes may act not only as CO_2_ fixation centers, but also participate in other physiological processes. The spatial distribution of carboxysomes in β-cyanobacteria is achieved through the operation of a two-component system, McdAB, that presumably employs a Brownian-ratchet mechanism to position these microbodies [193,194]. McdAB-like systems are also found in α-cyanobacteria [195].

The structural organization of carboxysomes undergoes significant changes depending on the level of C_i_ supply to a cell. Under C_i_ excess, pro-carboxysomes are found in the cytoplasm of β-cyanobacteria. They represent Rubisco and CA aggregates assembled together with CcmM [196,197,198]. Upon a decrease in C_i_ in the environment, these conglomerates become coated with a protein shell and are converted into carboxysomes. The acquisition of the shell with its selective properties allows these microparticles to become the full-fledged participants of the CCM.

### 4.5. The Model of Cyanobacterial CCM

The general scheme of the CCM operation in cyanobacteria is shown in Figure 4. As the CCM enters the high-affinity state, active C_i_ accumulation by the cell begins with the participation of three HCO_3_^−^ transporters (BCT1, SbtA and BicA) and two CO_2_ uptake systems (NDH-1_3_ and NDH-1_4_) (Section 4.2). The structure of the C_i_-uptake complex is significantly influenced by the habitats of each particular cyanobacterial species. Its composition, as well as the affinity of C_i_-uptake systems to their substrates, and their intrinsic flux rate, strictly corresponds to the needs of the species to ensure an inflow of C_i_ sufficient to saturate photosynthesis.

Regardless of the type of exogenous substrate, C_i_ enter a cell is accumulated as a pool of HCO_3_^−^ in the cytoplasm with a slightly alkaline pH around 7.4, which significantly reduces the risk of the spontaneous leakage of CO_2_. The concentration of HCO_3_^−^ in the cytoplasm of cells growing under a carbon limitation reaches 20–40 mM [71,74,199,200].

The final stage of CO_2_ concentration in cyanobacteria is carried out in carboxysomes. HCO_3_^−^ and RuBP enter this microcompartment through pores formed by the shell proteins [181]. Conversion of accumulated HCO_3_^−^ into CO_2_ molecules occurs with the participation of the carboxysomal CAs (CsoSCA, CcaA or CcmM) (Section 4.3), under presumably acidic pH values (<6.3) of the carboxysomal matrix [86]. The produced CO_2_ is used by Rubisco to carboxylate RuBP. The synthesized 3-PGA backflows from the carboxysome to the cytoplasm, where the rest of the RPP cycle reactions take place. The co-localization of Rubisco and CA in carboxysomes significantly reduces the leakage of CO_2_ molecules. The additional minimization of CO_2_ leakage with the simultaneous reduction of photorespiration is achieved due to the selective properties of the protein shell of carboxysomes. Limited amounts of CO_2_ leaked from carboxysomes are converted into HCO_3_^−^ by NDH-1_3/4_ complexes that act as intracellular C_i_ conversion systems along with a participation in CO_2_ uptake (Section 4.2). Through the work of NDH-1_3/4_, CO_2_ molecules are returned to the total cytoplasmic pool of C_i_.

## 5. CCM of *Chlamydomonas reinhardtii*

The CCM of microalgae is more complicated than that of cyanobacteria due to a larger number of subcellular compartments in the eukaryotic cell. Its structure and principles of functioning have been best studied in the model laboratory strain of the unicellular green microalga, *Chlamydomonas reinhardtii*. The effect of intracellular C_i_ accumulation under limiting concentrations of exogenous CO_2_ was also first discovered in *C. reinhardtii* [46]. The CCM organization of other eukaryotic microalgae is much less studied. The available information concerns mainly the individual components presumably involved in C_i_ concentration or some particular aspects of this process. A notable exception is the marine diatom algae, for which the CCM architecture and functioning was reconstructed using the whole genome sequence; and it appeared to be quite similar to the CCM of *C. reinhardtii* [53].

The CCM of *C. reinhardtii* requires at least 17 components to function properly [9,28,201,202]. These include nine systems for C_i_ transport across cell membranes (HLA3, LCI1, LCIA, CCP1, CCP2, CIA8 and BST1–3); three CAs (CAH1, CAH3 and CAH6); two peripheral pyrenoid proteins (LCIB and LCIC), presumably also possessing the CA activity; a methyltransferase homolog protein, CIA6; and two nuclear transcriptional regulators (CIA5 and LCR1). The exact function and importance of these individual participants is still not fully understood. The unambiguous necessity for providing directional C_i_ flow from the external environment to Rubisco is demonstrated for the following five proteins: HLA3, LCI1, LCIA, CAH3 and LCIB.

### 5.1. CCM: Induction and Two Photosynthetic Acclimation States of C. reinhardtii at Low C_i_

In *C. reinhardtii*, three physiological states of cells are distinguished during their acclimation to various CO_2_ concentrations in the environment [9,10]:L-cells (low-CO_2_ cells) are acclimated to low CO_2_ (0.04–0.5%). A total of 0.04% corresponds to the natural content of CO_2_ in the modern atmosphere.VL-cells (very-low-CO_2_ cells) are acclimated to very low (<0.02%) CO_2_ concentrations.H-cells (high-CO_2_ cells) are acclimated to elevated CO_2_ concentrations (2–5%).

Different sets of CCM components are responsible for the L- and VL-cells’ adaptation, resulting in their different photosynthetic and physiological properties [9]. Mainly, it concerns VL-cells with a higher affinity to CO_2_ and simultaneous decrease in the photosynthetic activity. In comparison to L-cells, VL-cells are distinguished by their small size, slow growth rate and low chlorophyll content. Despite such differences in physiological characteristics, L- and VL-cells displayed no differences in the level of the transcription of CCM-associated genes [203], pointing to some non-transcriptional switch that regulates the acclimation.

The mechanisms underpinning the microalgal CCM activation and transition to a high-affinity state have been significantly less studied than those in cyanobacteria. To date, it is evident that the CCM activity in *C. reinhardtii* does not depends only on the CO_2_ content in the external environment [22]. As with cyanobacteria (Section 4.1), it is assumed that a decrease in CO_2_ is perceived by microalgal cells via indirect signals, the appearance of which is associated with an imbalance between the light (photosynthetic electron transport) and dark (RPP cycle) phases of photosynthesis. Such signals include photorespiratory metabolites or changes in the redox status of cells. These are involved in retrograde signaling via specific mediators, such as CAS or GUN4, and serve as effectors for CCM induction [22].

The majority of the genes encoding the confirmed CCM components are practically not transcribed under elevated CO_2_, showing an increase in the expression level only under C_i_-limiting conditions. The exceptions are CAH3 and LCIB, which show low and moderate expression, respectively, under high CO_2_ with little or high increase in mRNA levels under carbon limitation ([203], supplementary data; [204], supplementary data). This may indicate that the corresponding proteins, the thylakoid CA CAH3 and the LCIB protein, which is a part of the C_i_ conversion system in the chloroplast stroma, are required for the successful growth of *C. reinhardtii* in a wide range of exogenous C_i_ concentrations.

The most complete picture of the *C. reinhardtii* cellular response to C_i_ limitation has been obtained only with the development of transcriptomics methods. A comprehensive analysis [203,204] revealed that at the beginning of the adaptation to the new environment, there is a global decrease in photosynthesis, a decrease in protein synthesis and in cell energy level, with a concurrent increase in photorespiration. An induction in the expression of the genes encoding the CCM components was observed after a short (~30 min) period.

CIA5 (also known as CCM1) and LCR1 are two CCM-associated regulatory proteins that have been identified in *C. reinhardtii*. The activity of other potential regulatory elements that could participate in the transcriptional regulation of the CCM activity is currently questionable [22]. CIA5 (C_i_ accumulation protein 5) is the major regulator of the CCM expression [205,206,207,208]. CIA5 controls all confirmed components of the CCM (Table 1 and Table 2), as well as a second regulatory protein, LCR1 (low CO_2_ stress response) [209]. Thus, the LCR1-regulated genes are ultimately controlled by CIA5. CIA5 is constitutively expressed regardless of the exogenic CO_2_ concentration, and it requires a post-translational activation under C_i_-limiting conditions [206,207,208]. CIA5 may function as a TF, or it can exert a regulatory function by modifying or interacting with other regulatory proteins [9,22]. LCR1 functions as a traditional TF [209] activating transcription with the participation of CIA5.

At the post-transcriptional level, the modulation of CCM activity in *C. reinhardtii* is accomplished by varying the translation efficiency via small regulatory RNAs [22] or post-translational modifications [9]. The role of phosphorylation in the activation of algal CCM components is well established. Phosphorylation affects both the CCM components and their immediate protein environment. The phosphorylation of the linker protein EPYC1 (LCI5), which is critical for pyrenoid assembly, regulates its interaction with Rubisco by altering the availability of protein-binding domains [102,210]. The phosphorylation of the lumenal CAH3 leads to a significant increase in its CA activity and migration from the stromal thylakoids to the pyrenoid tubules [67]. Diffusion of the LCIB/LCIC complex (a stromal C_i_ conversion system) toward the pyrenoid at extremely low CO_2_ [211,212] is apparently also associated with the phosphorylation of its protein components [213]. The global analysis of the *C. reinhardtii* phosphoproteome revealed that numerous other CCM-related proteins, such as HLA3, LCIC and several CAs (CAH6-9 and CAG2), were phosphorylated [214]. Another way of the post-translational modification is glutathionylation, which has been demonstrated for the LCIB and proteins involved in the RPP cycle [215]. This may indicate the role of oxidative stress and changes in the redox state of the cell under conditions of C_i_ starvation in the CCM regulation.

### 5.2. Transport of C_i_ in C. reinhardtii

Microalgae, like cyanobacteria, use both CO_2_ molecules and bicarbonate ions from the aqueous environment. On its way to Rubisco, C_i_ must overcome the plasma membrane, the chloroplast envelope and the thylakoid membrane. *C. reinhardtii* cells have a variety of experimentally proven and potential C_i_ transport systems across membranes (Table 1). Until recently, the MITC11 (mitochondrial carrier protein), a homolog to mitochondrial carrier proteins, was also considered as a potential C_i_ transporter in the chloroplast of *C. reinhardtii* [8]. However, to date, this role of MITC11 has neither been confirmed nor disproved. The chloroplast envelope protein CemA encoded by the *ycf10* gene [216] was also excluded from the list of potential C_i_ transporters. CemA is now assumed to work as a Na^+^-dependent proton pump, which removes H^+^ from the stroma [9].

Notably, the transcription of many *C. reinhardtii* genes involved in the function of the C_i_ uptake systems (HLA3, LCI1, LCIA, CCP1/2, and BST1–3) is enhanced upon a decrease in CO_2_. These genes are directly, or via LCR1, controlled by CIA5, the master regulator of the CCM genes’ expression [202,203,217] (Table 1). The involvement in the CCM-associated C_i_ uptake is now fairly well confirmed for HLA3, LCI1 and LCIA proteins. Impairments in the expression of the CIA8 and BST1–3 genes reduce the growth of cells and cause a decrease in their affinity to C_i_ under the CO_2_ limitation [201,202]. However, the activities of CIA8 and BST1–3 as the HCO_3_^−^ transporters still require the experimental confirmation. The transport of C_i_ across mitochondrial membranes with the help of potential transporters, CCP1 and CCP2, is thought to be important in coordinating CCM with photorespiration [10,22].

#### 5.2.1. Transport of C_i_ through the Plasma Membrane

*C. reinhardtii* has several proteins that have been proven or are suspected of being involved in C_i_ transport through the first barrier to the cell, the plasma membrane.

HLA3 (high light-activated protein) belongs to the family of ABC transporters with an ATP-binding cassette [208,218,219]. The *HLA3* was originally described as the gene induced by high-light intensity. The location of HLA3 in the plasma membrane is well documented [28]. Several works are devoted to elucidating its function [220,221,222]. The ability of HLA3 to transport HCO_3_^−^ has been unequivocally demonstrated in vitro in the *Xenopus* oocytes model system [28]. HLA3 is important for the adaptation of VL-cells and is inhibited by low CO_2_ in L-cells [223]. The enhanced transcription of the *HLA3* gene at low CO_2_ occurs through the CAS (calcium sensing) protein involved in the Ca^2+^-dependent retrograde signaling [224,225]. Recently, the SAGA1 protein, whose function was previously attributed only to the starch sheath formation and maintenance of the pyrenoid structure [104], was demonstrated to play the important role in this process [106].

LCI1 (low-CO_2_-inducible protein) was first identified as a plasma membrane protein that was expressed only at reduced CO_2_ concentrations [226,227]. LCI1 ensures an active CO_2_ uptake in L-cells, whereas its role in VL-cells is minimal [228]. Structural studies of LCI1 suggest that it may function as a gated plasma membrane CO_2_ channel [229].

In the plasma membrane of *C. reinhardtii*, HLA3 and LCI1 form a joint complex interacting with several other proteins [102,222]. In particular, both HLA3 and LCI1 interact with ACA4, which is a putative H^+^-exporting ATPase. It has been suggested that ACA4 may contribute to HCO_3_^−^ uptake, either by maintaining a proton gradient or by generating local cytosolic alkaline regions [102].

Two Rhesus-like proteins, RHP1 and RHP2, are found in the plasma membrane of *C. reinhardtii* and are thought to be potential CO_2_ channels [230,231,232]. The available information is mostly attributed to RHP1. The enhancement of the *RHP1* gene expression and the appearance of the corresponding protein in cells occurs only at high CO_2_ [203,233]. Earlier studies [217] supposed that the RHP1 expression is not controlled by CIA5. Other studies, however, suggest the involvement of CIA5 in the regulation of RHP1 [203]. Although RHP1 and RHP2 are not directly appointed to CCM, they may be important for the *Chlamydomonas* vital activity under elevated CO_2_ concentrations [233].

#### 5.2.2. Transport of C_i_ through the Chloroplast Envelope

The chloroplast envelope of *C. reinhardtii* forms a second barrier for the C_i_ transport toward Rubisco. Even recently, three possible candidates that may be associated with the HCO_3_^−^ transport into the chloroplast have been considered: the LCIA protein and the CCP1/CCP2 proteins [9]. Finally, CCP1/CCP2 were allocated to mitochondria [28], although they still retain their former acronyms (CCP, chloroplast carrier proteins).

The LCIA (low-CO_2_-inducible protein A) protein was originally annotated as a putative nitrite transporter based on its structural similarity to the formate/nitrite transporters (FNTs) [208,234]. This is why LCIA often appears under another name, NAR1.2 (nitrite assimilation-related). The ability of LCIA to assimilate both HCO_3_^−^ and NO_2_^−^ has been confirmed [28,234]. LCIA, like other bacterial FNTs, it is thought to act as a channel facilitating HCO_3_^−^ entry into the *Chlamydomonas* chloroplast by collaborating with the plasma membrane protein HLA3 [28,102,220,222].

Studies show that LCIA is physiologically relevant in VL-cells (0.01% CO_2_), and it is inhibited by ambient CO_2_ (0.04–0.5%) levels [204,223]. The activation of *LCIA* gene transcription, as well as that of *HLA3*, occurs at conditions of C_i_ deficiency and is mediated by the CAS protein [224].

The presence of a specialized CO_2_ transporter in the chloroplast envelope has not yet been reported. Thus, CO_2_ molecules transported by LCI1 into the cytosol may further enter the chloroplast stroma through an unidentified transporter or by passive diffusion [228].

#### 5.2.3. Bicarbonate Transporters in Thylakoid Membranes

For a long time, one of the main missing links in the scheme of photosynthetic C_i_ assimilation in *C. reinhardtii* was the HCO_3_^−^ transporter(s) from the chloroplast stroma to the thylakoid lumen, which supply bicarbonate ions to CAH3. The existence of such a transporter was a key argument for approving the CAH3 involvement in the CO_2_ supply to Rubisco, as well as in the WOC stabilization [65,87,93]. Therefore, the recent discovery of these CCM components became an eagerly anticipated event.

The *CIA8* (C_i_ accumulation protein 8) gene encodes a transmembrane protein belonging to the sodium/bile acid symporter family (SBF), which may also participate in the bicarbonate transport [201]. The CIA8 protein possesses a leader peptide that may targets this protein to the thylakoid membrane or to the chloroplast envelope. CIA8, expressed in-frame as a GFP fusion in *C. reinhardtii* cells, was not associated with the chloroplast envelope, but rather dispersed throughout the organelle [201], suggesting the thylakoid location of CIA8.

The accumulation of CIA8 transcripts at very low (0.01%) CO_2_ concentrations suggested its involvement in CCM [201]. Under CO_2_-limiting conditions, the CIA8 mutant showed a reduced growth, decreased affinity for C_i_ and decreased photosynthetic oxygen evolution rate. However, *CIA8* expression is not regulated by the CCM master regulator, CIA5. The precise location of CIA8 in *Chlamydomonas* cells, as well as the physiological role played by this protein, remain unknown.

Three bestrophin-like proteins, BST1, BST2, and BST3 (LCI11), represent additional potential bicarbonate transporters in *C. reinhardtii* thylakoid membranes [202]. These proteins interact with each other [102]. BST1–3 is found in both stromal and intrapyrenoid thylakoid membranes, with a marked concentration at the periphery of the pyrenoid [202]. It has been proposed that BST1–3, like a human bestrophin [235], functions as an anion channel. However, the ability of BST1–3 to transport HCO_3_^−^ needs to be confirmed.

The involvement of BST1–3 in the CCM of *C. reinhardtii* was suggested because *BST1*–*3* genes expression, being under the control of CIA5, strongly increased under low and very low CO_2_ [202,203,204]. Mutants with reduced BST1–3 expression were unable to grow at low CO_2_ exhibiting a reduced affinity to C_i_ and a reduced C_i_ uptake [202]. Interaction of BST1 and BST3 with the C_i_ conversion system in the stroma of chloroplasts and with the LCIB/LCIC complex [102] also suggested their involvement in CCM.

#### 5.2.4. Mitochondrial C_i_ Transporters

The mitochondrial membranes of *C. reinhardtii* contain CCP1 and CCP2 proteins [28], which belong to the MCP (mitochondrial carrier proteins) superfamily. Previously, CCP1 and CCP2 (chloroplast carrier proteins) were referred to as chloroplast membrane proteins, because they were originally found in the fraction of isolated chloroplasts [236,237,238].

CCP1 and CCP2 expression, which is under control of CIA5, significantly increased in VL-cells at extremely low CO_2_ [203,204]. Knock-outs deficient in CCP1 and CCP2 displayed growth retardation, whereas C_i_ accumulation and photosynthetic affinity for C_i_ were not affected [236].

It was suggested that CCP1 and CCP2 are involved in the transport of C_i_ from mitochondria to chloroplasts [22] during the acclimation to C_i_ deficiency. It is assumed that mitochondrial CAs (Section 5.3.3) participates in the conversion of CO_2_ released during respiration and photorespiration. The resulting HCO_3_^−^ can return to the chloroplast for recycling in the RPP cycle. This process should be especially important in VL-cells, allowing them to conserve the resources of intracellular C_i_.

### 5.3. Carbonic Anhydrases and C_i_ Conversion System in the Stroma of C. reinhardtii Chloroplast

The CA enzyme, which interconverts two forms of C_i_, CO_2_ and HCO_3_^−^, is the second important element of microalgal CCM. Fourteen genes were found in *C. reinhardtii* genome that encode the proven and potential CAs (Table 2). Three of these proteins belong to the α-class (CAH1, CAH2, and CAH3), six—to the β-class (CAH4-9), and three—to the γ-class of the enzyme (CAG1-3). Another two proteins are LCIB and LCIC, which probably belong to θ-class of CAs [82,213,239]. The CAs listed above have different locations in the cell and play different physiological roles [82,240] are discussed further below.

The master CCM regulator, CIA5, regulates the transcription of a number of *C. reinhardtii* CAs genes, which is enhanced under carbon-limiting conditions (Table 2). To date, only the following proteins have been experimentally attributed to the CCM: CAH1, CAH3, and the LCIB/LCIC complex [9,102,240]. It was suggested that other proteins (CAH2, CAH4, CAH5, CAH6, CAG1, CAG2, and CAG3) may indirectly participate in CCM.

#### 5.3.1. CAs of Periplasmic Space

The genome of *C. reinhardtii* includes two genes, *CAH1* and *CAH2* (carbonic anhydrase 1/2), that encode highly homologous α-CAs located in the periplasmic space [241,242,243]. Numerous studies reviewed in [9,240,244], show that CAH1 is involved in the maintenance of CCM functioning under low or very-low CO_2_. The function of CAH1 has been attributed to the delivery of HCO_3_^−^/CO_2_ to HLA3 and LCI1 transporters in the plasma membrane. Compared to CAH1, the CAH2 content in *C. reinhardtii* cells is rather low [244]. It should be noted that the *CAH2* gene is transcribed at a very low level, and this process is not regulated by CIA5 or reduced CO_2_ concentrations (Table 2). CAH2 was not thought to play a role in the CCM of *C. reinhardtii* and it is rather required for the assimilation of high CO_2_ concentrations [204,241].

In addition to α-class CAH1 and CAH2, two homologous β-class CAs, CAH7 and CAH8, were found in *C. reinhardtii* cells, and their specific activities were confirmed [245]. CAH8 is located in the periplasmic space, in close proximity to the plasma membrane [245]. If CAH8 is really associated with the membrane, it is still unknown which side of the membrane its active center is on; it was previously believed that it faces toward the periplasmic space [244]. Although the precise position of CAH7 has not been established, CAH8′s analogous extracellular location has been assumed [245]. There is ongoing debate over the CAH7/8 physiological functions, especially their role in CCM.

#### 5.3.2. Putative Cytosolic CA

The *CAH9* gene of *C. reinhardtii* encodes β-class CAs, the activity of which, however, has not been confirmed [85,244]. It should be noted that the name “CAH9” originally referred to the mitochondrial γ-class CA, now called CAG3 (GenBank IDs AAS48197 and AY463240). Nowadays, the name “CAH9” refers to the protein with GenBank ID ADW08083. This protein lacks a leader sequence, and it was localized in the microalgal cytoplasm [102]. *CAH9* is very weakly transcribed in *C. reinhardtii* cells at both low and high CO_2_ concentrations; the content of the corresponding protein in cells is also low [244]. Currently, this CA is not thought to play an important role in C_i_ accumulation and CO_2_ fixation [203,244].

#### 5.3.3. Mitochondrial CAs

Mitochondria of *C. reinhardtii* contain CAH4 and CAH5 proteins that belong to β-class CAs [102,246]. The *CAH4/5* transcription is strictly regulated by the CO_2_ concentrations [203,204,247]. Both proteins are found only in L-cells and are absent in H-cells [244].

The direct involvement of CAH4/5 in CCM is questionable [9]. These proteins, together with the mitochondrial bicarbonate transporters CCP1/2 (Section 5.2.4), may prevent CO_2_ (generated in mitochondria during respiration and photorespiration) leakage from cells, directing it for further fixation in the RPP cycle [248]. Involvement in such an economical intracellular C_i_ consumption may explain the accumulation of CAH4/5 under low CO_2_ conditions.

Additionally, three potential γ-class CAs, and CAG1, CAG2 and CAG3 (Glp1), have been found in *C. reinhardtii* mitochondria [102,249,250]. The *CAG1-3* genes are expressed constitutively, without any response to decreasing CO_2_ concentrations [204]. The CA activity was only assessed for the recombinant CAG3, but it was not confirmed [250]. If CAG1-3s do have enzymatic activity, their presumed function, like that of CAH4/5, would be to prevent the leakage of CO_2_ generated during respiration and photorespiration [240].

#### 5.3.4. Chloroplast CAs and C_i_ Conversion System

##### CAH3—Carbonic Anhydrase of the Thylakoid Lumen

α-class CA, CAH3, is located in the thylakoid lumen of *C. reinhardtii* [251]. The amount of *CAH3* mRNA increases slightly with a decrease in CO_2_ concentration [203,204]. *CAH3* expression level does not change visibly in the *CIA5* disrupted mutant of *C. reinhardtii*, suggesting that this CCM regulator is not involved in *CAH3* regulation ([203], supplementary data).

CAH3 is a constitutive protein that is required at both low and high CO_2_ concentrations [251]. CAH3 plays a key role in CCM: it supplies CO_2_ molecules to Rubisco, generating them from the HCO_3_^−^ in the lumen of interpyrenoid thylakoids [60,91,252,253]. In stromal thylakoids, where primary photosynthetic light reactions are carried out, CAH3 is bound to the polypeptides of the PSII reaction center [65]. CAH3 aids in the stabilization of the manganese cluster of PSII by hastening the removal of H^+^ released during the water-oxidizing reaction [92,93,94,254,255,256]. Thereby, CAH3 helps to facilitate the electron donation from the water to PSII and participates in the regulation of the light reaction of the photosynthesis.

##### Constitutive Complex LCIB/LCIC—Putative θ-Class CAs and C_i_ Conversion System in Chloroplast Stroma

Five proteins of the LCI family are encoded in the *C. reinhardtii* genome: LCIA, LCIB, LCIC, LCID and LCIE [9]. LCIA is a chloroplast envelope protein that transports HCO_3_^−^ from the cytosol to the plastid stroma (Section 5.2.2). In contrast to LCIA, LCIB–E proteins are soluble proteins and lack transmembrane domains. Among those, LCIB and LCIC are the most studied. Little information is available about LCID and LCIE: their subcellular localization and function, including the involvement in photosynthetic C_i_ assimilation and CCM, are currently unknown.

Crystallographic data revealed that LCIB and LCIC are homologues and structurally analogous to the active θ-class CA from the diatom alga *P. tricornutum* [213,239]. Therefore, LCIB and LCIC are regarded as potential θ-class CAs in *C. reinhardtii*; however, their specific enzymatic activity has yet to be confirmed.

Both proteins, LCIB and LCIC, have been found in the stroma of the *C. reinhardtii* chloroplast, where they form a heteromultimeric complex [102,211,212]. *LCIB* and *LCIC* are among the low CO_2_-induced, CIA5-regulated genes [203]. Notably, the LCIB/LCIC complex is also detected in H-cells, but its content is significantly increased in L- and VL-cells under CO_2_-limiting conditions. The complex is diffusely distributed in the stroma of L-cells, whereas it is concentrated near the pyrenoid in VL-cells [211,212,257]. The migration of the complex likely occurs with the assistance of LCIC and a starch sheath of the pyrenoid [258,259].

The involvement of LCIB/LCIC in the CCM of *C. reinhardtii* has been well documented. If LCIB and LCIC really possess the CA activity, the LCIB/LCIC complex should convert the CO_2_ entering the chloroplast into HCO_3_^−^, thereby maintaining a CO_2_ gradient for its facilitated uptake in both L- and VL-cells [9]. Such a function is similar to that of the cyanobacterial NDH-1_3/4_ systems (Section 4.2). In VL-cells, LCIB/LCIC can also act as a CO_2_ recapture system, preventing the CO_2_ leakage from the pyrenoid by its conversion into HCO_3_^−^ [212,257]. It is suggested that the CA reaction is carried out by LCIB, while LCIC serves as the integrating structural component [9].

#### 5.3.5. Flagellar CA

CAH6, an active β-class CA, was previously described as a chloroplast stromal enzyme in *C. reinhardtii* [260]. The recent in vivo analysis of the subcellular location of CAH6 using immunofluorescence microscopy allocated this protein to the flagella [102]. This finding confirmed the earlier data on flagella proteomes [261], as well as the whole proteome of *Chlamydomonas* chloroplast [262]. The CAH6 protein is found in both L- and H-cells [260]. CAH6 is slightly up-regulated in low CO_2_ environments, and its transcription is not regulated by the CIA5 [203]. CAH6 may be involved in the *Chlamydomonas* chemotaxis toward bicarbonate while determining the amount of C_i_ in the environment [102].

### 5.4. Pyrenoids

The Rubisco-containing microcompartment in microalgae is the pyrenoid, which serves as a functional analogue of cyanobacterial carboxysomes. Pyrenoids are protein complexes located in the chloroplasts of many eukaryotic microalgae [103,263,264]. These microbodies appear as electron-dense inclusions in the chloroplast, often surrounded by a starch sheath that becomes more pronounced under low CO_2_ (Figure 6a). The three-dimensional structure of pyrenoid, its protein composition and function, as well as the mutual arrangement of the CCM components associated with this microcompartment in *C. reinhardtii,* are well studied [101,102,103,105,265,266].

The proteomic analysis of the *C. reinhardtii* pyrenoid demonstrated that, in addition to Rubisco, it contains about 200 individual proteins [266]. Among them are pyrenoid matrix proteins, which are necessary for the packaging and efficient operation of Rubisco. These include Rubisco activase (chaperone protein RCA1), which ensures the carboxylation reaction at the highest possible rate [95], and the structural protein EPYC1 (essential pyrenoid component), which serves as a linker to promote the Rubisco aggregation in the pyrenoid [267] (previously known as LCI5—low-CO_2_-inducible protein 5). The presence of a variety of enzymes unrelated to the CO_2_ and starch metabolism in the *C. reinhardtii* pyrenoids suggests that these microcompartments may conduct previously unknown functions. Notably, the putative methyltransferase CIA6, which was attributed to the pyrenoid assembly [268], was not reported in the pyrenoid proteome ([266], supplementary data). Simultaneously, the intracellular location of CIA6 in *Chlamydomonas* cells is uncertain [102].

Thylakoid stacks in the stroma and the pyrenoid are connected by cylindrical pyrenoid tubules, which pass between starch grains and enter the pyrenoid matrix [265] (Figure 6b). Each pyrenoid tubule of *C. reinhardtii* contains 2–8 parallel minitubules inside, which are formed when two or more thylakoids merge into a common large tubule. Thus, the internal content of the minitubules is filled with chloroplast stroma; this solves the problem of the apparent physical isolation of Rubisco from other enzymes of the RPP cycle. The lumen of pyrenoid tubules is connected with the lumen of stromal thylakoids, allowing coordination of the chloroplasts’ light and dark reactions (Figure 7).

It was suggested that HCO_3_^−^ enters inside the thylakoids from the stroma side via CIA8 and BST1–3 transporters [201,202] and further diffuses inside the pyrenoid through the lumen of large pyrenoid tubules [265]. Alternatively, bicarbonate can diffuse to the interior of the pyrenoid directly via minitubules or enter into the lumen of pyrenoid tubules via its transporters. The minitubules also carry the unidirectional diffusion of ATP molecules required for Rubisco activase activity and exchange of RuBP and 3-PGA, which are substrates and products of the Rubisco carboxylase activity [95,264,269].

CAH3, which supplies CO_2_ to Rubisco, is located in the lumen of large pyrenoid tubules [60,92,250,251] (Figure 7). CAH3 is crucial for cell growth at low CO_2_ levels. Under these conditions, CAH3 is phosphorylated and transferred from stromal thylakoids to pyrenoid tubules [67]. Simultaneously, phosphorylation leads to a significant increase in the CA activity of CAH3. In the lumen of the pyrenoid thylakoid tubules, HCO_3_^−^ is converted to CO_2_ by CAH3 (Figure 7). This reaction is facilitated by the high concentration of H^+^ in the lumen of illuminated cells, which shifts the CO_2_/HCO_3_^−^ interconversion reaction towards the CO_2_ generation. The resulting lipophilic CO_2_ molecules can diffuse from the thylakoids into the pyrenoid matrix, where they are fixed by Rubisco in the RPP cycle. It should be highlighted that contrary to long-held beliefs, the pyrenoid matrix acts more like a liquid than a crystal-like or amorphous solid structure [101,270]. This knowledge may aid in imagining how the CO_2_ substrate penetrates into Rubisco’s active centers, despite its apparent density.

The pyrenoid is thought to spatially separate Rubisco from the oxygen-evolving complex because the thylakoid membranes of the pyrenoids are enriched in active PSI complexes and devoid of PSII [95,266]. This arrangement results in a low O_2_/CO_2_ ratio that favors the carboxylase function to Rubisco and reduces its oxygenase activity. Although fluorescently labeled components of PSI and PSII, cytochrome *b*_6_/*f*-complex and ATP synthase have been detected within *C. reinhardtii* pyrenoids [102], these results contradict with the proteomic data obtained on the isolated pyrenoids [266].

Pyrenoids are often surrounded by a starch sheath whose thickness increases during the acclimation to low CO_2_ [58]. The pyrenoid matrix and starch sheath bind together through the protein SAGA1 (starch granules abnormal 1), which also regulates the starch sheath morphology [104]. It is believed that the starch sheath can serve as a barrier, preventing CO_2_ leakage from the pyrenoid. However, surprisingly, the absence of the sheath does not interfere with the CCM activity [66,74,271]. To date, the barrier role of the starch sheath has not either been confirmed or disproved. However, its shape (morphology) has been demonstrated to be closely tied to the CCM’s functioning [104] and to the correct localization of the LCIB protein [259]. In addition, the pyrenoid-localized protein SAGA1 is necessary for the CAS-dependent expression of *HLA3* and *LCIA* genes, encoding C_i_ transporters [106].

### 5.5. The Models of C. reinhardtii CCM under Varying Availability of C_i_

The currently accepted scheme of CCM in microalgal cells, in general, does not substantially differ from the initial models [87,91,252,272,273,274]. However, it is now much better resolved due to the information acquired over more than 40 years of research regarding its molecular components, subcellular location and functions. Figure 8 shows a scheme of photosynthetic C_i_ assimilation under a different CO_2_ supply in *C. reinhardtii*.

Figure 8a shows the photosynthetic CO_2_ fixation pathway in H-cells grown at elevated CO_2_ concentrations (2–5%) in the absence of the active C_i_ uptake. Under these conditions, the photosynthesis is ensured by the following CCM components: CAs CAH2 and CAH3, CO_2_-channels RHP1/2, and the LCIB/LCIC complex. CO_2_ can enter the cell by a direct diffusion through cell membranes. The CO_2_ transport channels (RHP1/2) located in the plasma membrane may exert their functions with the participation of CAH2 [241]. At elevated CO_2_ concentrations, the pyrenoid structures are weak, the starch sheath is absent and Rubisco is located mainly in the stroma. Thus, the CO_2_ entering the cell is fixed in the RPP cycle directly in the chloroplast stroma. At the same time, the LCIB/LCIC complex can convert the CO_2_ entering the cell into HCO_3_^−^; this process is facilitated by slightly alkaline stromal pH values. Under the C_i_ excess, the main physiological role of the lumenal CA of thylakoids, CAH3, is obviously not in generation of CO_2_ for Rubisco, but in the protection of WOC of PSII from the excess of protons generated due to the activity of the electron transport chain [94]. HCO_3_^−^ molecules enter into the lumen of stromal thylakoids via BST1–3 and CIA8 transporters. CAH3 catalyzes the interaction of HCO_3_^−^ with protons to form CO_2_, which diffuses according to a concentration gradient into the stroma, where it can be consumed for RuBP carboxylation. The constitutive expression of LCIB and CAH3 proteins under high CO_2_ allows cells to respond immediately to the CO_2_ shortage, avoiding the delay associated with the de novo synthesis of the complete set of CCM proteins [9].

Figure 8b,c shows the organization of the CCM in the L- and VL-cells of *C. reinhardtii* adapted to low (0.03–0.5%) and very low (<0.02%) CO_2_, respectively. It is assumed that the high CO_2_ concentration near the Rubisco active centers under these conditions is achieved, as follows.

(1)CAH6, located in the flagella of *C. reinhardtii* [102], may estimate an ambient C_i_ concentration, providing the orientation of cells relative to its gradient and enabling the directional chemotaxis (not shown in Figure 8).(2)C_i_ enters into the cell and further into the chloroplast by active transport in two forms, HCO_3_^−^ and CO_2_. The periplasmic CA, CAH1, evidently contributes to the interconversion of C_i_ forms and participates in the supply of substrates for the plasma membrane transporters. The transport of HCO_3_^−^ through the plasma membrane and chloroplast envelope is facilitated by the cooperative operation of HLA3 and LCIA [28,102,222]. It appears that the active HCO_3_^−^ uptake occurs only in VL-cells, whereas in L-cells, the HLA3 and LCIA transporters are inhibited by CO_2_ [223]. The CO_2_ uptake in both L- and VL-cells may be due to its facilitated diffusion, driven by light-dependent stromal alkalinization and the operation of the LCIB/LCIC complex [9,257]. In L-cells, the flow of CO_2_ inside the cell is additionally provided by the LCI1 channel [228]. CO_2_ can enter the chloroplast not only by direct diffusion, but also through some, as of yet, unidentified transporter. The absorbed C_i_ accumulates in the chloroplast stroma in the form of HCO_3_^−^, which is facilitated by its slightly alkaline pH values.(3)Stromal HCO_3_^−^ enters inside the thylakoids with the participation of CIA8 and BST1–3 transporters located in membranes on the stroma side, followed by a diffusion deep into the pyrenoid along the lumen of large pyrenoid tubules [201,202]. Simultaneously, HCO_3_^−^ can diffuse inside the pyrenoid along the stroma inside the minitubules and then enter the lumen of intrapyrenoid thylakoids via transporters. The conversion of HCO_3_^−^ into CO_2_ in the lumen of intrapyrenoid thylakoids is performed by CAH3, which exploits the inherently acidic pH value of this compartment [9,60,85,92].(4)CO_2_ formed in the intrapyrenoid thylakoid lumen diffuses into the pyrenoid matrix, where it is fixed by Rubisco in the RuBP carboxylation reaction. The co-localization of Rubisco and CA in the pyrenoid is the cornerstone of the CCM scheme, necessary for the efficient incorporation of CO_2_ into the initial chain of organic carbon transformations in the cell.(5)The prevention of CO_2_ leakage from the chloroplast is also very important. The current CCM model suggests that this process is primarily important in VL-cells [9,10]. This function is performed by the LCIB/LCIC protein complex (C_i_ conversion system) located in the chloroplast stroma. In VL-cells, the complex migrates to the periphery of the pyrenoid and acts as a vector module for the conversion of CO_2_ that escaped from the pyrenoid into HCO_3_^−^ [211] (Figure 8c). This assumption is based on the high probability that LCIB/LCIC possess the CA activity. Under low CO_2_, the main function of the LCIB/LCIC complex is probably to hydrate the incoming cellular CO_2_ with the formation of HCO_3_^−^ in the stroma [212].(6)It is assumed that mitochondrial CAs and C_i_ transporters may participate, although indirectly, in the CCM of *C. reinhardtii* [22,244,248] (not shown in Figure 8). The CO_2_ generated during photorespiration and respiration in these organelles can be converted by CAH4 and CAH5 into HCO_3_^−^, with the latter released into the cytosol via CCP1/2 transporters. Then, HCO_3_^−^ can be returned to the chloroplast and reintroduced into the dark phase of photosynthesis.

## 6. Conclusions

The CCM of microalgae and cyanobacteria has been actively studied over the past 40 years after its discovery in the 1980s. The most significant progress in the study of its cellular and molecular organization (especially in microalgae), and understanding the subtleties of this mechanism functioning and regulation, have been made in the last decade due to the development of *omics* methodology and structural analysis. Many aspects of the CCM organization and regulation are being revised and evolved on a constant basis. Yet, some critical parts and characteristics of CCM operation remain unexplored. The following are the primary “hot spots” that warrant more investigation:More nuanced understanding of the mechanisms of CCM regulation. In particular, it remains largely unclear how its light regulation is performed;In-depth study of the functioning of the CA system and C_i_ transport;Determination of the mechanisms by which CO_2_ enters the algal chloroplast;Confirmation of the CA activity of the LCIB/LCIC complex in algae and CupA/B proteins in cyanobacteria;Evaluation of external, mitochondrial, and flagella CAs for their participation in photosynthetic assimilation of C_i_ and CCM;More direct experimental measurements of pH values in the periplasmic space of microalgae and cyanobacterial cells, as well as the pH of the carboxysomal matrix in cyanobacteria, would aid in understanding the role of CAs in C_i_ flow control;Clarification of the involvement of CIA6 protein in pyrenoid biogenesis, and the exact role of the starch sheath in the CCM;Currently, the CCM has been studied in detail only in model laboratory species of cyanobacteria and microalgae. However, the sets of CCM components and the significance of this mechanism may differ in species inhabiting different ecological niches;In addition, special attention should be paid to the study of the evolutionary origin of CCM in interrelation with the geological history of Earth and the history of the biosphere.

## Figures and Tables

**Figure 1 plants-12-01569-f001:**
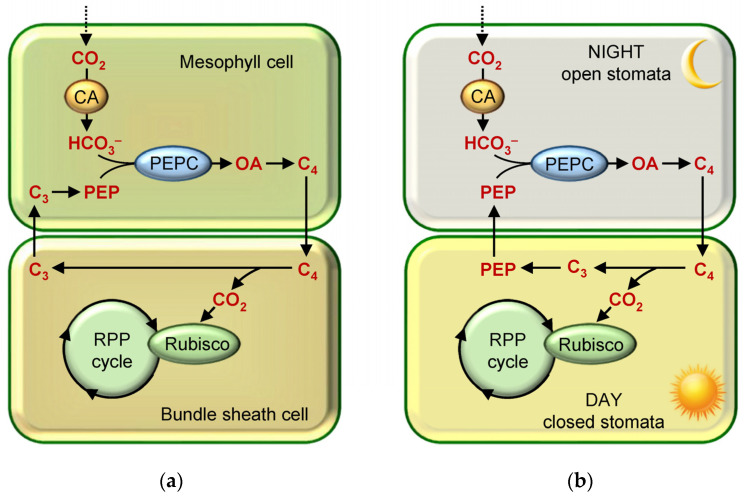
Schematic representation of dark phase photosynthesis reactions in (**a**) C_4_ plant cells; (**b**) CAM plants. Abbreviations: C_3_/C_4_—C_3_/C_4_-dicarboxylic acids; OA—oxaloacetate; PEP—phosphoenolpyruvate; PEPC—phosphoenolpyruvate carboxylase.

**Figure 2 plants-12-01569-f002:**
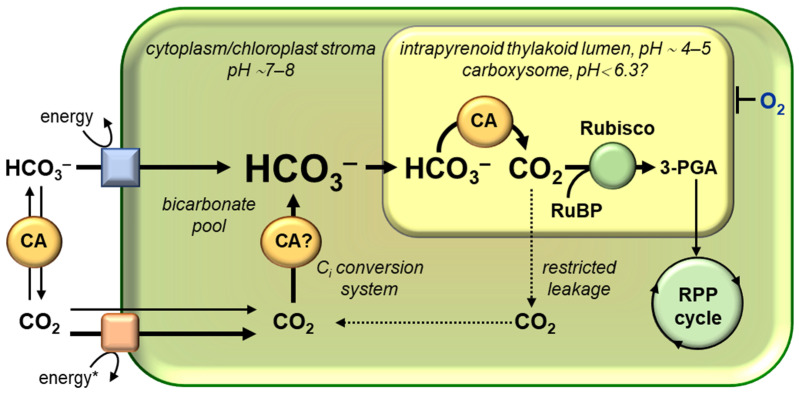
Schematic representation of the CCM in microalgal and cyanobacterial cells. The font size reflects the relative concentrations of CO_2_ and HCO_3_^−^ in the external environment and inside the cell. Thick arrows show the main direction of C_i_ flow. * Energy requirements for the promoted CO_2_ uptake are currently approved only for cyanobacteria.

**Figure 3 plants-12-01569-f003:**
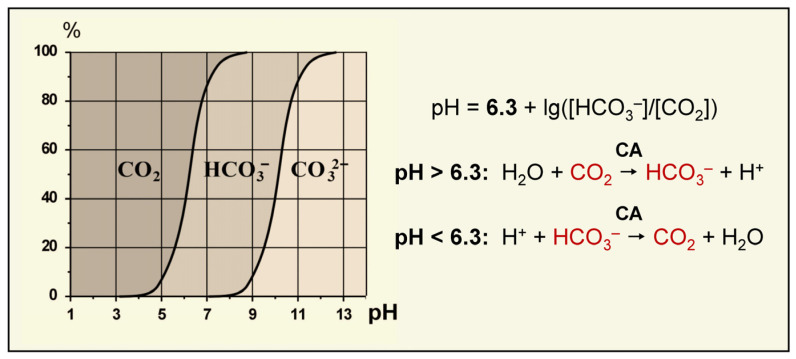
The concentration ratio and pH dependence of C_i_ forms in water solutions [65]; the Henderson–Hasselbach equation for [CO_2_]/[HCO_3_^−^] ratio; and the preferential direction of the reaction ensured by CA as a function of pH.

**Figure 4 plants-12-01569-f004:**
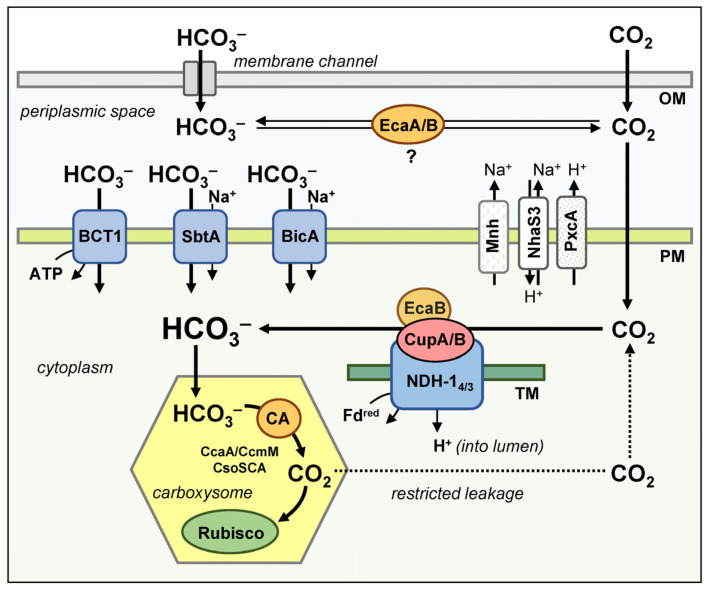
CCM in model strains of cyanobacteria. The periplasmic CAs, EcaA and EcaB are also shown, but their involvement in the CCM has not yet been confirmed. Abbreviations: OM—outer membrane; PM—plasma membrane; TM—thylakoid membrane.

**Figure 5 plants-12-01569-f005:**
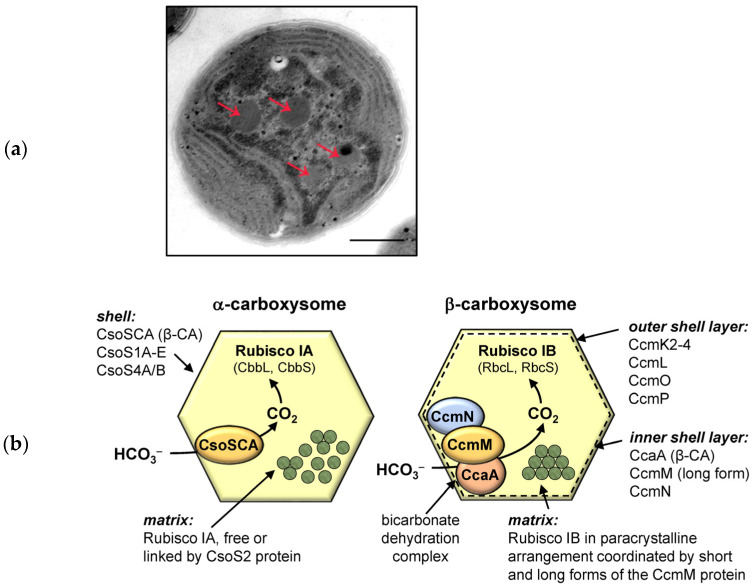
Cyanobacterial carboxysomes. (**a**) Carboxysomes (arrows) in the cytoplasm of the cyanobacterium *Synechocystis* sp. strain PCC 6803. Image was kindly provided by Dr. M.A. Sinetova (K.A. Timiryazev Institute of Plant Physiology, RAS, Moscow). Scale bar is 0.5 µm. (**b**) Structural components of α- and β-carboxysomes in *Prochlorococcus marinus* MED4 (α-cyanobacteria) and *Synechococcus elongatus* PCC 7942 (β-cyanobacteria). Modified from [174]; supplemented from [89,175]. The proteins of the inner layer of the β-carboxysome shell form a bicarbonate-dehydrating complex that converts HCO_3_^−^ to CO_2_ for the RPP cycle [184]. The organizing link of the complex is a full-length form of the CcmM protein, which simultaneously binds Rubisco, CcmN and CcaA (β-CA). The complex is associated with the carboxysome shell through specific interactions of CcmM with proteins in outer shell layer, CcmK and CcmL. Bicarbonate dehydration is performed by the CcaA or CcmM CAs.

**Figure 6 plants-12-01569-f006:**
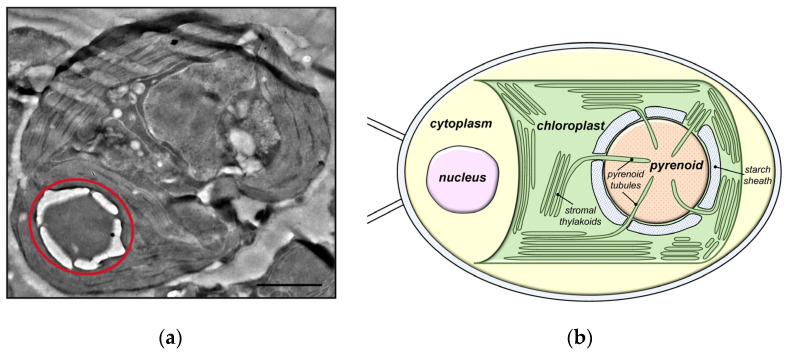
Pyrenoid in *C. reinhardtii* cells. (**a**) The TEM image showing the pyrenoid inside the chloroplast of *Chlamydomonas* was kindly provided by Dr. Maria A. Sinetova (K.A. Timiryazev Institute of Plant Physiology, RAS, Moscow). Scale bar is 1 µm. (**b**) Schematic cross-sectional view of a *Chlamydomonas* cell showing the branching of cylindrical pyrenoid tubules from the thylakoid stacks and their entrance into the pyrenoid matrix through fenestrations in the starch sheath.

**Figure 7 plants-12-01569-f007:**
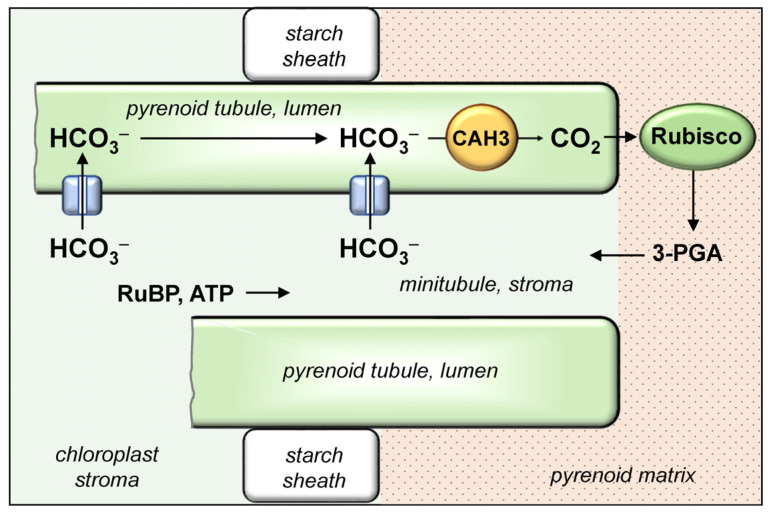
Pyrenoid tubules with minitubules: schematic representation of the cross-sectional plane of *C. reinhardtii* cells at the entrance of pyrenoid tubules into the pyrenoid matrix.

**Figure 8 plants-12-01569-f008:**
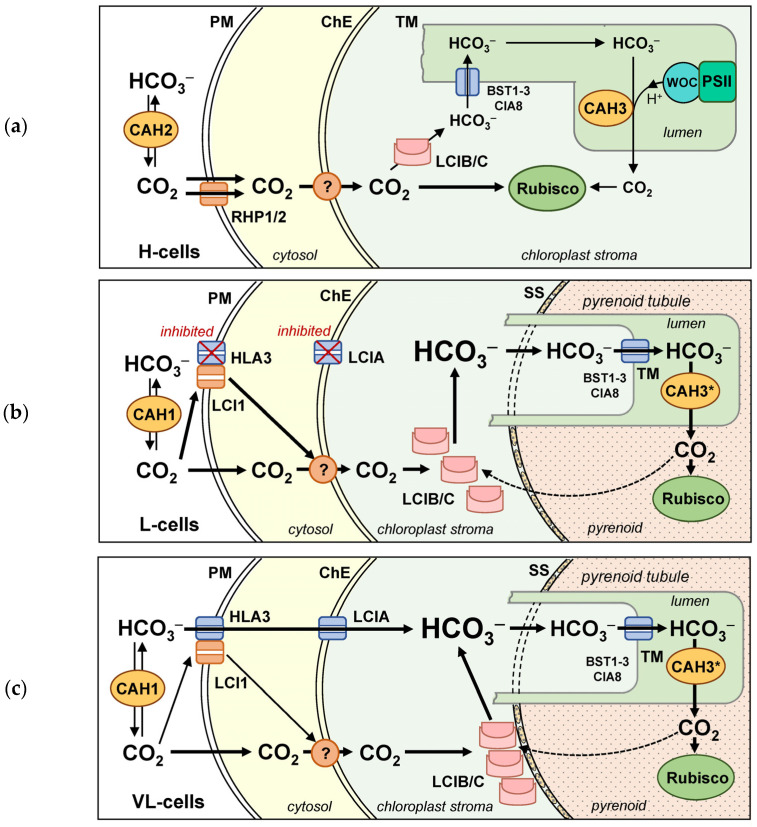
Schematic of photosynthetic C_i_ assimilation in *C. reinhardtii* under different CO_2_ supply. (**a**) H-cells growing under conditions of elevated CO_2_ concentrations (2–5%), with inactive CCM; Generalized model of CCM functioning in *C. reinhardtii* cells grown at: (**b**) low (0.03–0.5%; L cells) and (**c**) very low (<0.02%; VL cells) CO_2_ concentrations. The scheme shows bicarbonate entry into the lumen of intrapyrenoid thylakoids (pyrenoid tubules) from minitubules in L- and VL-cells, and does not take into account the participation of transporters located on membranes on the stroma side. Abbreviations: ChE—chloroplast envelope; PM—plasma membrane; SS—starch sheath; TM—thylakoid membrane; CAH3*, activated enzyme. The preferential pathways of C_i_ fluxes in the corresponding acclimation state of the cell are shown in bold.

**Table 1 plants-12-01569-t001:** Confirmed and potential C_i_ transport systems in *C. reinhardtii*.

Cellular Location	Protein	Function	Regulation of Gene Expression		Acclimation State
			Low CO_2_	CIA5	
Plasma membrane	**HLA3**	HCO_3_^−^ transport	+	+	VL-cells
	**LCI1**	CO_2_ uptake	+	+(via LCR1)	L-cells
	RHP1	CO_2_ uptake	−	+	H-cells
	RHP2	CO_2_ uptake	−	n/d	H-cells
Chloroplast envelope	**LCIA**	HCO_3_^−^ transport	+	+	VL-cells
Thylakoid membrane	**CIA8**	HCO_3_^−^ transport (?)	+	−	H-, L-, VL-cells
	**BST1**	HCO_3_^−^ transport (?)	+	+	H-, L-, VL-cells
	**BST2**	HCO_3_^−^ transport (?)	+	+	H-, L-, VL-cells
	**BST3**	HCO_3_^−^ transport (?)	+	+	H-, L-, VL-cells
Mitochondria	**CCP1**	HCO_3_^−^ transport (?)	+	+	VL-cells
	**CCP2**	HCO_3_^−^ transport (?)	+	+	VL-cells

Components with confirmed or proposal involvement in the CCM are shown in bold. n/d—no data.

**Table 2 plants-12-01569-t002:** Carbonic anhydrases and C_i_ conversion system with putative CA activity in *C. reinhardtii*.

Location	Protein	CA Class	Activity	Physiological Role **	Regulation of Gene Expression		Acclimation State
					Low CO_2_	CIA5	
Periplasmic space	**CAH1**	α	+	Supply C_i_ for uptake	+	+ (via LCR1)	L-, VL-cells
	CAH2	α	+	Supply C_i_ for uptake	−	−	H-cells
Periplasmic space or plasma membrane	CAH7	β	+	n/d	−	−	n/o
	CAH8	β	+	n/d	−	+	n/o
Cytosol	CAH9	β	n/d	n/d	−	+	n/o
Mitochondria	CAH4	β	+	Conversion CO_2_ into HCO_3_^−^	+	+	L-, VL-cells
	CAH5	β	+	Conversion CO_2_ into HCO_3_^−^	+	+	L-, VL-cells
	CAG1	γ	n/d	Conversion CO_2_ into HCO_3_^−^	−	n/d	n/o
	CAG2	γ	n/d	Conversion CO_2_ into HCO_3_^−^	−	n/d	n/o
	CAG3	γ	−	Conversion CO_2_ into HCO_3_^−^	−	n/d	n/o
Chloroplast stroma	**LCIB** *	θ	−	CO_2_ leakage prevention	+	+	H-, L-, VL-cells
	**LCIC** *	θ	−	CO_2_ leakage prevention	+	+	H-, L-, VL-cells
Thylakoid lumen	**CAH3**	α	+	CO_2_ generation for Rubisco	±	n/o	H, L-, VL-cells
Flagella	CAH6	β	+	C_i_ sensing	−	−	n/o

Components with confirmed or proposed involvement in the CCM are shown in bold. * Proteins LCIB and LCIC form a complex that functions as the C_i_ conversion system in the chloroplast stroma. ** known or predicted. n/d—no data. n/o—not obviously.

## Data Availability

Not applicable.

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
