# Peer review of "Adapting from Low to High: An Update to CO2-Concentrating Mechanisms of Cyanobacteria and Microalgae"

_plants, 2023, doi:10.3390/plants12071569_

Round 1

Reviewer 1 Report

No comments. 

Reviewer 2 Report

This MS focused on the induction of CCM, the adaptation mechanism of low Ci, the migration mechanism of Ci, and the cooperative role of CA in CCM, and reviewed the research progress of CCM in cyanobacteria and Chlamydomonas reinhardtii. This review is overall well-presented. I only have some minor issues that the author would consider.

1. Research on Chlamydomonas reinhardtii is extensive, and the CCM of microalgae in this paper is Chlamydomonas reinhardtii as an example, so it is suggested to add Chlamydomonas reinhardtii to the keywords.

2. As mentioned in Section 4.2, line 554, "BicA is the only Ci transport system usually detected in marine cyanobacteria.", it is suggested to add supporting or illustrative content.

3. In the section of abstract, it is suggested to appropriately add relevant research background of CCM to make the abstract more clear to read.

4. “In microalgae, the appearance of CCM was, most likely, independent.” I think this sentence may be replaced by "was" by "is"  

Reviewer 3 Report

The paper is interesting and well written in all aspects, including also the taxonomic ones, usually not adequately reported. There are several sentences which could be open to discussion, but the method adopted by the authors was a strict reference to the current literature. I would only suggest some possible inputs. Line 54, I suggest to change the sentence "Aquatic photosynthetic organisms, namely microalgae and cyanobacteria," in "Aquatic photosynthetic microorganisms, namely microalgae and cyanobacteria," to avoid any confusion with aquatic plants like Posidonia. Please,  also reconsider some sentences, like those in lines 824-825, 661-665, and 233-234, being the matter in continuous evolution. For instance, about the photosynthetic process in lines 233-234, dark phase should considered integrated part of the intere photosynthetic process and not independent, as correctly tested few lines further. This aspect, concerning the evolution of data, experiments, interpretations and approaches could be also reported in the Conclusion.
